# CRYOFM: A FLOW-BASED FOUNDATION MODEL FOR CRYO-EM DENSITIES

**Yi Zhou**[*], **Yilai Li**[*], **Jing Yuan**[*], **Quanquan Gu**[†]
ByteDance Research
{zhouyi.naive,yilai.li,yuanjing.eugene,quanquan.gu}@bytedance.com

## ABSTRACT

Cryo-electron microscopy (cryo-EM) is a powerful technique in structural biology and drug discovery, enabling the study of biomolecules at high resolution. Significant advancements by structural biologists using cryo-EM have led to the production of around 40k protein density maps at various resolutions on EMDB[1]. However, cryo-EM data processing algorithms have yet to fully benefit from our knowledge of biomolecular density maps, with only a few recent models being data-driven but limited to specific tasks. In this study, we present CRYOFM, a foundation model designed as a generative model, learning the distribution of high-quality density maps and generalizing effectively to downstream tasks. Built on flow matching, CRYOFM is trained to accurately capture the prior distribution of biomolecular density maps. Furthermore, we introduce a flow posterior sampling method that leverages CRYOFM as a flexible prior for several downstream tasks in cryo-EM and cryo-electron tomography (cryo-ET) without the need for fine-tuning, achieving state-of-the-art performance on most tasks and demonstrating its potential as a foundational model for broader applications in these fields.

## 1 INTRODUCTION

Cryo-electron microscopy is an important technique in structural biology and drug discovery, allowing the determination of high-resolution 3D structures of biomolecules that are difficult to study through conventional methods, offering insights into molecular mechanisms and aiding in fields like drug discovery (Nogales & Scheres, 2015). A major component of cryo-EM data processing involves reconstructing 3D structures from noisy 2D projections of particles. It can be formulated as an inverse problem (Bendory et al., 2020; Singer & Sigworth, 2020), which aims at recovering the signal $\mathbf{x} \in \mathbb{R}^n$ (i.e., a clean protein density) from the observation $\mathbf{y} \in \mathbb{R}^m$ [2]. Under this formulation, following Bayesian statistics, the objective is to sample a density from the posterior $p(\mathbf{x}|\mathbf{y})$, which can be factorized to $p(\mathbf{x}|\mathbf{y}) \propto p(\mathbf{y}|\mathbf{x})p(\mathbf{x})$. Thus, the prior distribution becomes essential for guiding the reconstruction process and improving the accuracy of the resulting structures.

Scheres (2012b) first introduced a Gaussian distribution as the prior $p(\mathbf{x})$ in 3D reconstruction for cryo-EM, effectively functioning as a frequency-dependent low-pass filter, which has proven useful in many cases. Building on this, recent works have explored more sophisticated regularizers rather than explicitly defining a prior distribution (Punjani et al., 2020; Tegunov et al., 2021; Kimanius et al., 2021; Li et al., 2024; Schwab et al., 2024; Kimanius et al., 2024; Liu et al., 2024). While these approaches were initially developed for 3D reconstruction tasks, their methodologies are closely aligned with the methods for density map modification and post-processing (Jakobi et al., 2017; Ramírez-Aportela et al., 2020; Terwilliger et al., 2020a; Kaur et al., 2021; Sanchez-Garcia et al., 2021; He et al., 2023). Both share the common objective of improving the quality of cryo-EM maps, whether during refinement or in post-processing. Among all these approaches, a line of methods has emerged that utilizes data-driven models to directly learn $p(\mathbf{x}|\mathbf{y})$ from data, showing strong potential (Sanchez-Garcia et al., 2021; He et al., 2023; Kimanius et al., 2024). While powerful, these models are often designed for specific tasks, limiting their general applicability and versatility.

---

[*]Equal contribution, [†]Corresponding author.
[1]EMDB (wwPDB Consortium, 2023) is a public repository for cryo-EM volumes.
[2]The signal $\mathbf{x} \in \mathbb{R}^{D \times D \times D}$ denotes a clean protein density, where $D$ is the side length. The observation $\mathbf{y}$ can have any arbitary shape, we use $\mathbf{x} \in \mathbb{R}^n$ for ease of notation but without loss of generality.

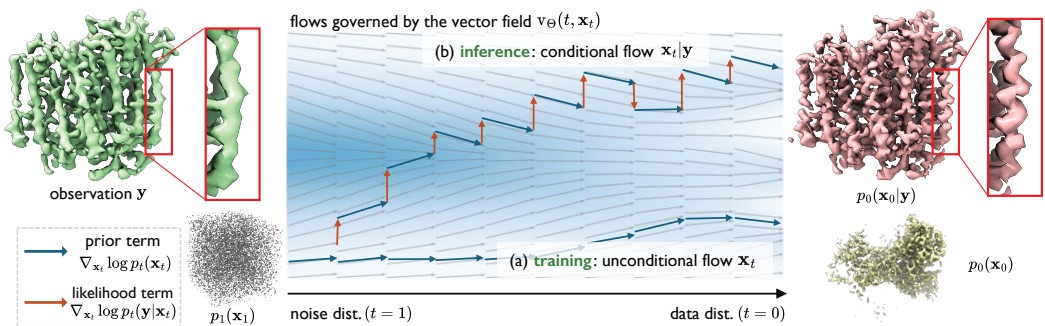

Figure 1: The overview of CRYOFM. In the training stage, CRYOFM learns a vector field $v_\Theta(t, \mathbf{x}_t)$, whose corresponding probability flow generates the data distribution $p_0(\mathbf{x}_0)$ of high-quality protein densities. In the inference stage, given an observation $\mathbf{y}$, a likelihood term $p_t(\mathbf{y}|\mathbf{x}_t)$ is incorporated to convert the unconditional vector field $v_\Theta(t, \mathbf{x}_t)$ to a conditional one $v_\Theta(t, \mathbf{x}_t|\mathbf{y})$, so that we can sample from the posterior distribution $p_0(\mathbf{x}_0|\mathbf{y})$. This enables signal restoration of the density map, resulting in improved resolution of the alpha helices in the shown case.

Foundation models leverage prior knowledge and provide versatility across a wide range of applications. In particular, they have revolutionized AI for protein structure prediction and design by utilizing vast datasets to tackle diverse tasks (Jumper et al., 2021; Baek et al., 2021; Lin et al., 2023; Krishna et al., 2024; Hayes et al., 2024; Abramson et al., 2024; Wang et al., 2024b). Chroma (Ingraham et al., 2023) and RFDiffusion (Watson et al., 2023) are two foundation models for protein structure modeling, which are broadly applicable to many structure-related tasks. However, experimental electron density maps, a key modality in structural biology, have largely been neglected in this space. Meanwhile, the cryo-EM field has not yet taken full advantage of foundation models, which integrate prior knowledge and offer flexibility across multiple applications. Wang et al. (2024a) developed a condition diffusion model in the density space, which achieves superior results by refining the map for model building, but is limited to single downstream task. In this study, we present a *foundation model*, CRYOFM, that directly models the prior distribution $p(\mathbf{x})$ by learning the distribution of high-quality cryo-EM density maps through flow matching. Notably, flow matching (Lipman et al., 2022; Liu et al., 2022a) is a well-established generative model, which achieves state-of-the-art results in many applications (Dao et al., 2023; Esser et al., 2024; Yim et al., 2023a). The key ingredient of flow matching is a time-dependent vector field $v_\Theta(t, \mathbf{x}_t)$, which can be employed to progressively denoise a noise vector into a data sample and is closely related to the score of the prior distribution $\nabla_{\mathbf{x}_t} \log p(\mathbf{x}_t)$. In CRYOFM, when applied to downstream tasks, we draw a sample from the posterior distribution with its score (Ho et al., 2020; Song et al., 2021b): $\nabla_{\mathbf{x}} \log p(\mathbf{x}|\mathbf{y}) = \nabla_{\mathbf{x}} \log p(\mathbf{x}) + \nabla_{\mathbf{x}} \log p(\mathbf{y}|\mathbf{x})$. In contrast to expert-designed prior, a data-driven prior is more powerful and expressive. Compared with models that directly learn the posterior, a decomposition of the prior and likelihood is more robust and versatile. As long as we can derive the likelihood term for a downstream task, we can combine it with the learned prior in a plug-and-play fashion (Zhang et al., 2021; Venkatakrishnan et al., 2013) to sample from the posterior.

CRYOFM consists of two models: CRYOFM-S, which captures fine local details at high resolution, and CRYOFM-L, which focuses on the overall global shape of the biomolecule at medium to low resolution[3]. Fig. 1 provides an overview of CRYOFM. In the training stage, we regress the vector field with high-quality density data downloaded from the EMDB database (wwPDB Consortium, 2023). When applied to downstream tasks, we propose a flow posterior sampling method to convert the vector field $v_\Theta(t, \mathbf{x}_t)$ to a conditional one $v_\Theta(t, \mathbf{x}_t|\mathbf{y})$, which can be used to sample from the posterior distribution $p(\mathbf{x}|\mathbf{y})$. When applied to various downstream tasks, CRYOFM achieves state-of-the-art results on most of the experiments, demonstrating the power and potential of CRYOFM as a *foundation model* for solving more complex problems in cryo-EM and cryo-ET.

Our main contributions are summarized as follows: (1) we present the first flow-based generative model as a foundation model that learns the distribution of high-quality cryo-EM density maps; (2) we derive a flow posterior sampling algorithm, enabling CRYOFM to be effectively utilized as a prior for various downstream tasks; (3) in an unsupervised manner, we achieve better performance in several downstream tasks compared to DeepEMhancer (Sanchez-Garcia et al., 2021), EMReady

---

[3]Unless stated otherwise, CRYOFM refers to CRYOFM-S throughout this paper.

(He et al., 2023) and spIsoNet (Liu et al., 2024); (4) we explore different model architectures and configurations to optimize the training of generative models on large biomolecular density maps.

## 2 RELATED WORK

In this section, we will briefly review the mostly related work on cryo-EM and diffusion/flow matching for inverse problems. A more detailed discussion on these topics can be found in Section B.

**Density modification and denoising in cryo-EM** Density modification refers to the process of using known properties of the expected density in specific regions of a map to correct errors in observed cryo-EM density maps (Terwilliger et al., 2020b). Deep learning has increasingly contributed to cryo-EM map denoising and modification, with methods divided into two categories: pretrained models and self-supervised approaches. Pretrained models, like DeepEMhancer (Sanchez-Garcia et al., 2021) and EMReady (He et al., 2023), learn the posterior $p(\mathbf{x}|\mathbf{y})$ from data to recover high-frequency details. In contrast, self-supervised approaches, such as M (Tegunov et al., 2021) and spIsoNet (Liu et al., 2024), are trained on the dataset being processed, offering more robustness but take longer to process one dataset.

**Missing wedge problem in cryo-ET** Cryo-electron tomography (cryo-ET) is an imaging technique used to reconstruct 3D volumes of biological specimens from 2D images captured at various tilt angles (Lučić et al., 2005). The resulting 3D volumes, known as tomograms, provide detailed views of cellular structures, with smaller regions, called subtomograms, extracted to focus on specific structures like proteins (Wan & Briggs, 2016). A key challenge in cryo-ET is the missing wedge problem, caused by the limited range of tilt angles during data acquisition, which leaves a wedge-shaped region in Fourier space without information, leading to anisotropic resolution and artifacts. Traditional approaches use signal processing and regularization techniques to address the missing wedge (Goris et al., 2012; Deng et al., 2016; Yan et al., 2019; Zhai et al., 2020), while recent deep learning methods have shown promise in tackling this issue by leveraging data-driven models (Liu et al., 2022b; Van Veen et al., 2024).

*Ab initio* **modeling in cryo-EM** *Ab initio* modeling in cryo-EM involves estimating the 3D structure of a protein from 2D particle images with unknown orientations (Crowther et al., 1970). One of the early approaches involved using 2D class averages, which are representative images created by aligning and averaging particles with similar views to improve the signal-to-noise ratio (SNR) (Ludtke et al., 1999; Voss et al., 2010). However, since the introduction of cryoSPARC, modern methods now use raw particle images directly for model estimation, leveraging stochastic gradient descent (SGD) to bypass the need for class averages (Punjani et al., 2017).

**Diffusion/Flow Matching for Inverse Problem** Inverse problems refer to a class of problems where the goal is to recover the original sample from the forward model's observation, such as image super-resolution (Haris et al., 2018), inpainting (Yeh et al., 2017), deblurring (Kupyn et al., 2019), etc. Since a pretrained DDPM/Flow models the data distribution (prior distribution), it can help discover the posterior distribution given a forward model (likelihood). Specifically, Chung et al. (2022) developed diffusion posterior sampling for general inverse problems, removing the need for strong assumptions like the linearity of the forward operator. Another line of research examines the solution to the inverse problem when the forward operator is unknown (Chung et al., 2023a; Kapon et al., 2024). For protein structure modeling, Levy et al. (2024) adopts Chroma (Ingraham et al., 2023) as the diffusion prior for a dozen of downstream tasks, including atomic model refinement in cryo-EM.

## 3 PRELIMINARIES

In this section, we briefly review two key components of cryoFM. In the training stage, flow matching (Section 3.1) is employed as the framework to learn the distribution $p(\text{density})$. In the inference stage, given some observations (noisy densities, 2D projections, etc.), we use the posterior sampling (Section 3.2) technique to sample a density from $p(\text{density}|\text{observation})$.

### 3.1 FLOW MATCHING

Flow matching (Lipman et al., 2022; Liu et al., 2022a) defines the generative process of a data point $\mathbf{x}_0 \in \mathbb{R}^n$ by continuously transforming a sample $\mathbf{x}_1 \in \mathbb{R}^n$ from the noise distribution, where the

state at an intermediate timestep $t \in [0, 1]$ is denoted by $\mathbf{x}_t \in \mathbb{R}^n$. A *flow* is the path traversed by the state $\mathbf{x}_t$, which is characterized by an ordinary differential equation (ODE):

$$d\mathbf{x}_t = \mathbf{v}_\Theta(t, \mathbf{x}_t)dt, \tag{1}$$

where $\mathbf{v}_\Theta : [0, 1] \times \mathbb{R}^n \to \mathbb{R}^n$ is a *time-dependent vector field* parameterized by a neural network parameterized by $\Theta$, which defines the transformation of the state $\mathbf{x}_t$. At each time step $t$, the state $\mathbf{x}_t$ follows a distribution $p_t(\mathbf{x}_t)$, where $p_t : [0, 1] \times \mathbb{R}^n \to \mathbb{R}_{\geq 0}$ is a time-dependent probability density function, i.e., $\int p_t(\mathbf{x}_t)d\mathbf{x}_t = 1$. Since $p_t(\mathbf{x}_t)$ evolves from the noise distribution $p_1(\mathbf{x}_1)$ to the data distribution $p_0(\mathbf{x}_0)$, it is referred to as a *probability density path* generated by the vector field $\mathbf{v}_\Theta(t, \mathbf{x}_t)$. An illustration of the concepts of flow matching can be found in Fig. 12.

Given a set of data points sampled from $p_1(\mathbf{x}_1)$ and $p_0(\mathbf{x}_0)$, flow matching aims to learn the vector field $\mathbf{v}_\Theta$ from the data. Chen et al. (2018) directly solves Eq. (1) via a differentiable neural ODE solver. Lipman et al. (2022) and Liu et al. (2022a) suggest that it is more efficient to learn a manually designed conditional vector field $\mathbf{u}_t(\cdot|\mathbf{x}_0) : [0, 1] \times \mathbb{R}^n \to \mathbb{R}^n$ given a data point $\mathbf{x}_0$. For example, the conditional flow can be a straight line between $\mathbf{x}_0$ and a noise sample $\mathbf{x}_1$:

$$\mathbf{x}_t|\mathbf{x}_0 = (1 - t)\mathbf{x}_0 + t\mathbf{x}_1, \tag{2}$$

where the corresponding vector field is $\mathbf{u}_t(\mathbf{x}_t|\mathbf{x}_0) = \mathbf{x}_1 - \mathbf{x}_0$. Finally, the conditional flow matching objective (Lipman et al., 2022) is defined as follows:

$$\mathcal{L}(\Theta) = \mathbb{E}_{t\sim\mathcal{U}[0,1],\mathbf{x}_0\sim p_0(\mathbf{x}_0),\mathbf{x}_1\sim p_1(\mathbf{x}_1)}\|\mathbf{v}_\Theta(t, \mathbf{x}_t) - (\mathbf{x}_1 - \mathbf{x}_0)\|_2^2,$$

which is equivalent to optimizing the original flow matching objective (Lipman et al., 2022).

## 3.2 DIFFUSION POSTERIOR SAMPLING

Diffusion posterior sampling (DPS) has emerged as a promising approach for solving inverse problems (Song et al., 2021b;a; Chung et al., 2023b). Given a measurement $\mathbf{y} \in \mathbb{R}^m$ derived from $\mathbf{x} \in \mathbb{R}^n$ with a forward operator $\mathcal{A} : \mathbb{R}^n \to \mathbb{R}^m$, the goal is to sample $\mathbf{x}$ from the posterior $p(\mathbf{x}|\mathbf{y})$. By Bayes' theorem, it is easy to show that the score of the posterior distribution is:

$$\nabla_{\mathbf{x}_t} \log p_t(\mathbf{x}_t|\mathbf{y}) = \nabla_{\mathbf{x}_t} \log p_t(\mathbf{x}_t) + \nabla_{\mathbf{x}_t} \log p_t(\mathbf{y}|\mathbf{x}_t). \tag{3}$$

However, computing $\log p_t(\mathbf{y}|\mathbf{x}_t) = \log \int_{\mathbf{x}_0} p(\mathbf{x}_0|\mathbf{x}_t)p_0(\mathbf{y}|\mathbf{x}_0)d\mathbf{x}_0$ is intractable since it requires the integration over all possible $\mathbf{x}_0 \sim p_0(\mathbf{x}_0|\mathbf{x}_t)$. Chung et al. (2022) proposed to use a Laplace approximation of the likelihood term: $p_t(\mathbf{y}|\mathbf{x}_t) \approx p_0(\mathbf{y}|\hat{\mathbf{x}}_0(\mathbf{x}_t))$, where $\hat{\mathbf{x}}_0(\mathbf{x}_t)$ is the posterior mean estimated by the diffusion model. The conditional score can thus be approximated by:

$$\nabla_{\mathbf{x}_t} \log p_t(\mathbf{x}_t|\mathbf{y}) \approx \nabla_{\mathbf{x}_t} \log p_t(\mathbf{x}_t) + \nabla_{\mathbf{x}_t} \log p_0(\mathbf{y}|\hat{\mathbf{x}}_0(\mathbf{x}_t)),$$

If we assume the observation distribution $p_0(\mathbf{y}|\mathbf{x})$ is Gaussian conditioned on $\mathbf{x}$, we can show that the gradient of the log-likelihood satisfies:

$$\nabla_{\mathbf{x}_t} \log p_0(\mathbf{y}|\hat{\mathbf{x}}_0(\mathbf{x}_t)) \propto -\nabla_{\mathbf{x}_t}\|\mathbf{y} - \mathcal{A}\hat{\mathbf{x}}_0(\mathbf{x}_t)\|_2^2. \tag{4}$$

Putting all these things together, we can approximate the score of the posterior distribution by:

$$\nabla_{\mathbf{x}_t} \log p(\mathbf{x}_t|\mathbf{y}) \approx \nabla_{\mathbf{x}_t} \log p_t(\mathbf{x}_t) - \lambda_t \cdot \nabla_{\mathbf{x}_t}\|\mathbf{y} - \mathcal{A}\hat{\mathbf{x}}_0(\mathbf{x}_t)\|_2^2,$$

where $\lambda_t$ is associated with the partition function of Gaussian, and we treat it as a hyperparameter to control the step size of the likelihood term.

## 4 CRYOFM

In this section, we present the implementation of CRYOFM. First, we introduce the pretraining dataset in Section 4.1. Next, we illustrate the architecture of the neural network in Section 4.2. Finally, we propose a posterior sampling algorithm for the downstream tasks.

### 4.1 PRETRAINING DATASET

Our training dataset consists of deposited sharpened density maps from the EMDB (wwPDB Consortium, 2023), specifically those with: 1) a reported resolution better than 3.0 Å, 2) structures resolved by single-particle cryo-EM, and 3) data entries that include half-maps, ensuring that resolution estimates are based on the "gold standard" Fourier shell correlation (FSC) (Henderson et al.,

2012). We manually curated this subset by removing exceptionally large complexes, helical structures, and problematic cases through visual inspection (see Fig. 13 for examples). We also excluded density maps with side lengths greater than 576 Å. This curation resulted in a total of 3479 density maps, where 32 density maps were selected as test set and excluded from training. The density maps in the selected subset were lowpass filtered to 1.5 Å/voxel and 3 Å/voxel for CRYOFM-S and CRYOFM-L, respectively. For CRYOFM-S, we applied random cropping to volumes of size $64^3$ along with random rotations for augmentation, whereas data for CRYOFM-L were center cropped to volumes of size $128^3$ and augmented solely with random rotations. Additionally, we rescale the density value to avoid large numerical variance (Section C.2 describes the details).

## 4.2 ARCHITECTURE

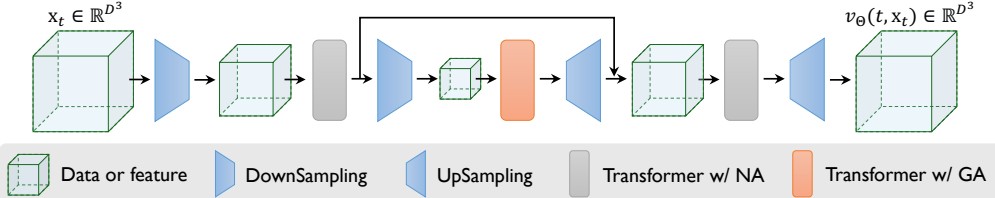

Figure 2: CRYOFM's architecture. The side length $D$ of the input $\mathbf{x}_t$ undergoes dimension reduction through down-sampling layers and is then expanded back to its original size. To minimize computational cost near the input and output, the model employs neighborhood attention (NA). Neighborhood attention only attend to a localized area, whereas global attention (GA) calculates attention across all positions.

A major challenge in applying the vanilla Transformer architecture to 3D density $\mathbf{x}_t \in \mathbb{R}^{D^3}$ is the computational complexity, which is $\mathcal{O}(D^6)$ [4]. CRYOFM processes the 3D input data through a Transformer with a *hierarchical* architecture, based on HDiT (Crowson et al., 2024). As illustrated in Fig. 2, the hierarchical structure downsamples the spatial dimension at initial levels and upsamples it at final levels, implemented via PixelUnshuffle and PixelShuffle layers (Shi et al., 2016). Consequently, the spatial dimension at the middle level is significantly reduced, making the model more easily scalable. Furthermore, at the two ends of the hierarchical structure, CRYOFM employs Neighborhood Attention (NA) (Hassani et al., 2023) since a localized attention layer is able to capture local dependencies while considerably reducing the number of tokens to attend.

## 4.3 FLOW POSTERIOR SAMPLING

Given a vector field $\mathbf{v}_\Theta(t, \mathbf{x}_t)$ that generates the prior data distribution $p_0(\mathbf{x}_0)$, we aim to convert it to a vector field $\mathbf{v}_\Theta(t, \mathbf{x}_t|\mathbf{y})$ that generates the posterior $p_0(\mathbf{x}_0|\mathbf{y})$. First, according to Dao et al. (2023); Song et al. (2021b), we can connect the vector field with the score function $\nabla_{\mathbf{x}_t} \log p_t(\mathbf{x}_t)$ as follows:

$$\mathbf{v}_\Theta(t, \mathbf{x}_t) = f_t(\mathbf{x}_t) - \frac{g_t^2}{2} \nabla_{\mathbf{x}_t} \log p_t(\mathbf{x}_t),$$

where $f_t$ is the drift term and $g_t$ is the diffusion coefficient in the forward-time SDE. In addition, we define the conditional vector field as follows:

$$\mathbf{v}_\Theta(t, \mathbf{x}_t|\mathbf{y}) = f_t(\mathbf{x}_t) - \frac{g_t^2}{2} \nabla_{\mathbf{x}_t} \log p_t(\mathbf{x}_t|\mathbf{y}) = \mathbf{v}_\Theta(t, \mathbf{x}_t) - \frac{g_t^2}{2} \nabla_{\mathbf{x}_t} \log p_t(\mathbf{y}|\mathbf{x}_t),$$

where the second equality holds due to Eq. (3). Thus, substracting the score of the likelihood term weighted by the diffusion coefficient generates a vector field that models the posterior distribution. Following the rectified flow (Dao et al., 2023), we set $f_t(\mathbf{x}_t) = -\frac{\mathbf{x}_t}{1-t}$ and $\frac{g_t^2}{2} = \frac{t}{1-t}$ in the conditional flow in Eq. (2), and obtain the conditional vector field as follows:

$$\mathbf{v}_\Theta(t, \mathbf{x}_t|\mathbf{y}) = \mathbf{v}_\Theta(t, \mathbf{x}_t) - \frac{t}{1-t} \nabla_{\mathbf{x}_t} \log p_t(\mathbf{y}|\mathbf{x}_t). \tag{5}$$

Similar to Section 3.2, by substituting Eq. (4) into Eq. (5), we obtain:

$$\mathbf{v}_\Theta(t, \mathbf{x}_t|\mathbf{y}) \approx \mathbf{v}_\Theta(t, \mathbf{x}_t) + \frac{t}{1-t} \cdot \lambda_t \nabla_{\mathbf{x}_t} \|\mathbf{y} - \mathcal{A}\hat{\mathbf{x}}_0(\mathbf{x}_t)\|_2^2. \tag{6}$$

---

[4]The computational complexity of Transformers is $\mathcal{O}(L^2)$, where $L = D^3$ is the number of tokens.

---

**Algorithm 1** Flow Posterior Sampling

---

**Require:** a pretrained vector field $\mathbf{v}_\Theta : [0, 1] \times \mathbb{R}^n \to \mathbb{R}^n$, a forward operator $\mathcal{A} : \mathbb{R}^n \to \mathbb{R}^m$, an obseravation $\mathbf{y} \in \mathbb{R}^m$, number of steps $N$, maximum step size $\lambda_{\max}$ at each step
**Ensure:** the recovered signal $\mathbf{x}_0$
 $\Delta t \leftarrow \frac{1}{N}$
 $\mathbf{x}_1 \leftarrow \mathcal{N}(0, \boldsymbol{I})$
 **for** $t \in [1, 1 - \Delta t, 1 - 2\Delta t, \cdots, \Delta t]$ **do**
  $\mathbf{x}'_{t-\Delta t} \leftarrow \mathbf{x}_t - \Delta t \cdot \mathbf{v}_\Theta(t, \mathbf{x}_t)$
  $l(\mathbf{x}_t) \leftarrow \|\mathbf{y} - \mathcal{A}\hat{\mathbf{x}}_0(\mathbf{x}_t)\|_2^2$
  $\mathbf{g} \leftarrow \frac{\nabla_{\mathbf{x}_t} l(\mathbf{x}_t)}{\|\nabla_{\mathbf{x}_t} l(\mathbf{x}_t)\|_2}$         $\triangleright$ Normalize the gradient for numerical stability
  $\lambda'_t \leftarrow \min\left\{\lambda_{\max}, \frac{t}{1-t}\right\}$       $\triangleright$ Prevent the weighting term from being too large
  $\mathbf{x}_{t-\Delta t} \leftarrow \mathbf{x}'_{t-\Delta t} - \lambda'_t \mathbf{g}$
 **end for**
 **return** $\mathbf{x}_0$

---

where $\hat{\mathbf{x}}_0(\mathbf{x}_t) = \mathbf{x}_t - t \cdot \mathbf{v}_\Theta(t, \mathbf{x}_t)$ is the posterior mean estimated by the flow model. In practice, we incorporate some tricks to avoid numerical instability. The detailed algorithm[5] of flow posterior sampling is described in Alg. 1. The algorithm can be adapted for different tasks; we refer the reader to Appendix G for the additional versions.

## 5 EXPERIMENTS

Fourier Shell Correlation (FSC) is a widely used metric that compares two density maps in Fourier space (Harauz & van Heel, 1986), allowing for the assessment of alignment between the ground truth and the reconstructed map. In the experiments, to evaluate the quality of the density maps, we use three primary metrics: $\text{FSC}_{\text{AUC}}$, $\text{FSC}_{0.5}$, and Fail Rate (FR). Specifically, $\text{FSC}_{\text{AUC}}$ measures the overall correlation across all spatial frequencies, and $\text{FSC}_{0.5}$ represents the resolution of the reconstructed map at the standard 0.5 cutoff (Rosenthal & Henderson, 2003). The Fail Rate (FR) identifies cases where the method fails to run or produces a result that significantly deviates from the ground truth. For the FSC metrics, we only report the results for cases that did not fail. More detailed explanation can be found in Section E.1.

### 5.1 SPECTRAL NOISE DENOISING

In cryo-EM reconstruction, spectral noise is the most commonly used noise model due to its ability to capture the varying noise characteristics across different spatial frequencies, which arise from factors such as the contrast transfer function (CTF) and detector imperfections. We consider the introduction of noise in the Fourier domain, where the variance of the added Gaussian noise is frequency-dependent, with higher frequencies exhibiting greater noise variance. Given a density $\tilde{\mathbf{V}} \in \mathbb{C}^{D \times D \times D}$ in the Fourier domain, the forward model $\mathcal{A} : \mathbb{C}^{D \times D \times D} \to \mathbb{C}^{D \times D \times D}$ is:

$$\mathcal{A}(\tilde{\mathbf{V}}) = \tilde{\mathbf{V}} + \epsilon,$$

where $\epsilon \in \mathbb{R}^{D \times D \times D}$ is a noise volume, whose values on the spherical shell with the same radius $\nu$ are the same ($\nu$ denotes the index of a component in the frequency space):

$$\epsilon(\nu) \sim \mathcal{N}(0, \sigma_{\text{noise}}^2(\nu)). \tag{7}$$

In the experiment, we estimate $\sigma_{\text{noise}}^2(\nu)$ from the half maps, which are two independent observations with the same underlying signal with uncorrelated noise. See Section F.1 for details.

We manually introduce spectral noise, resulting in resolutions ranging from 3.2 Å to 15.0 Å. Deep-EMhancer and EMReady, both pretrained models that directly learn the posterior $p(\mathbf{x}|\mathbf{y})$, are used as baselines. CRYOFM demonstrates superior robustness, with no failed cases across all experiments, while both baselines show higher fail rates, especially at medium resolutions. Additionally, as shown in Tab. 1 and Tab. 5 in Section F.1, CRYOFM achieves higher $\text{FSC}_{\text{AUC}}$ and better $\text{FSC}_{0.5}$ in

---

[5]The step size $\lambda'_t$ in Alg. 1 is not equivalent to $\lambda_t$ in Eq. (6). This is because solving an ODE requires multiplying the velocity by the length of a time interval, i.e., $\lambda'_t = \lambda_t \cdot \frac{1}{\text{time-steps}}$.

Table 1: Results of the spectral noise denoising task, comparing CRYOFM with DeepEMhancer (Sanchez-Garcia et al., 2021) and EMReady (He et al., 2023). In Sections 5.1, 5.2, 5.3, the forward model acts as a degradation of the original data. The best attainable $FSC_{0.5}$ is 3.0 Å since the voxel size is 1.5 Å.

| | Estimated resolution at different level of noise | | | | | | | | |
| | 3.2 Å | | | 4.3 Å | | | 6.1 Å | | |
| | FR↓ | $FSC_{AUC}$ ↑ | $FSC_{0.5}$ ↓ | FR↓ | $FSC_{AUC}$ ↑ | $FSC_{0.5}$ ↓ | FR↓ | $FSC_{AUC}$ ↑ | $FSC_{0.5}$ ↓ |
|---|---|---|---|---|---|---|---|---|---|
| After degradation | - | 0.8876 | 3.21 | - | 0.5527 | 4.96 | - | 0.4969 | 6.07 |
| DeepEMhancer | 0.06 | $0.92 \pm 0.02$ | $\mathbf{3.0 \pm 0.0}$ | 0.34 | $0.66 \pm 0.05$ | $\mathbf{4.3 \pm 0.2}$ | 0.44 | $\mathbf{0.62 \pm 0.05}$ | $\mathbf{4.5 \pm 0.3}$ |
| EMReady | 0.03 | $0.73 \pm 0.02$ | $3.4 \pm 0.1$ | 0.16 | $0.59 \pm 0.08$ | $4.9 \pm 0.9$ | 0.19 | $0.55 \pm 0.09$ | $5.3 \pm 0.9$ |
| CRYOFM | $\mathbf{0.00}$ | $\mathbf{0.95 \pm 0.01}$ | $\mathbf{3.0 \pm 0.0}$ | $\mathbf{0.00}$ | $\mathbf{0.68 \pm 0.04}$ | $4.4 \pm 0.2$ | $\mathbf{0.00}$ | $\mathbf{0.62 \pm 0.04}$ | $4.6 \pm 0.3$ |

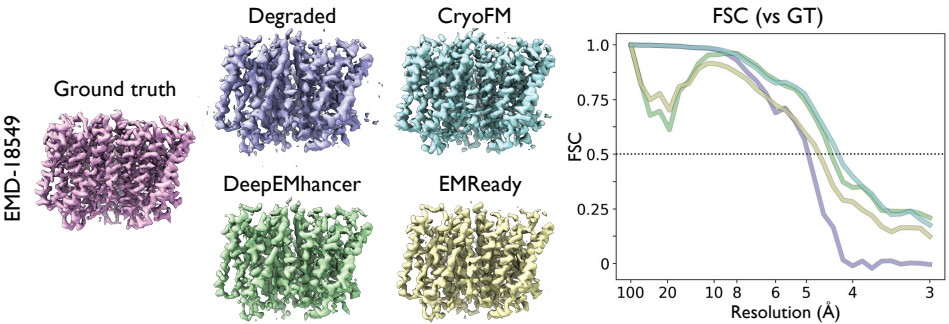

Figure 3: Result of the spectral noise denoising task. Two density maps from EMDB were added spectral noise so that the estimated resolution is 4.3 Å. The degraded density maps (results after applying the forward model) were filtered by *relion_postprocess* for visual clarity.

most cases, significantly improving degraded density maps and enhancing the resolution of helices and loops, as illustrated in Fig. 3. Notably, the FSC curves demonstrate that CRYOFM successfully improves the signal across all frequencies, while the baselines fails to retain trustworthy signals in the low-frequency range.

## 5.2 ANISOTROPIC NOISE DENOISING

Anisotropic noise in cryo-EM occurs when noise distribution varies by direction, affecting some orientations more than others and leading to uneven reconstruction quality. As a result, certain orientations have lower signal-to-noise ratios than others. To approximate this degradation in a simplified manner, we amplify the spectral noise by a factor when the particle orientation falls within a specific range of angles, simulating the increased uncertainty in less-sampled orientations. The forward model $\mathcal{A} : \mathbb{C}^{D \times D \times D} \to \mathbb{C}^{D \times D \times D}$ is given by:

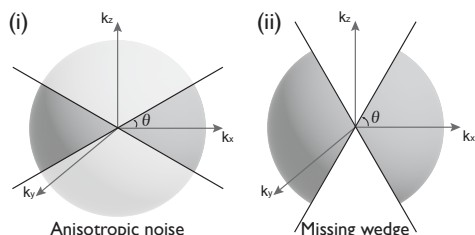

Figure 4: Forward operators for (i) the anisotropic noise and (ii) the missing wedge.

$$\mathcal{A}(\tilde{\mathbf{V}}) = \begin{cases} \tilde{\mathbf{V}} + \epsilon, & \text{if } \theta_{\min} \leq \theta \leq \theta_{\max} \\ \tilde{\mathbf{V}} + \alpha \cdot \epsilon, & \text{otherwise} \end{cases},$$

where $\epsilon \in \mathbb{R}^{D \times D \times D}$ is the same as defined in Eq. (7), and $\alpha > 1$ is a scalar that increases the noise for specific orientation angles, reflecting the heightened uncertainty in those directions. $\theta_{\min}$ and $\theta_{\max}$ are the angles that controls the portion of the noise being amplified as illustrated in Fig. 4. In the experiment, we estimate $\sigma^2_{\text{noise}}(\nu)$ and $\alpha$ from the half maps. See Section F.2 for details.

We manually add varied anisotropic spectral noise to the test set density maps and use Deep-EMhancer, spIsoNet, and CRYOFM for signal restoration. As shown in Tab. 2, CRYOFM outperforms both baselines on FSC metrics and has a zero fail rate. Fig. 5 demonstrates that CRYOFM restores the corrupted signal without introducing significant bias, unlike DeepEMhancer, which introduces artifacts and alters the overall shape. While spIsoNet preserves signals across frequencies, it fails to improve correlation with the ground truth. This suggests that pretrained models that learn the posterior $p(\mathbf{x}|\mathbf{y})$ like DeepEMhancer can struggle to retain reliable signals, and self-supervised

Table 2: Results of the anisotropic noise denoising task, comparing CRYOFM with DeepEMhancer (Sanchez-Garcia et al., 2021) and spIsoNet (Liu et al., 2024).

| | $\theta \in [-45°, +45°]$ | | | $\theta \in [-30°, +30°]$ | | | $\theta \in [-15°, +15°]$ | | |
| | FR↓ | $FSC_{AUC}$ ↑ | $FSC_{0.5}$ ↓ | FR↓ | $FSC_{AUC}$ ↑ | $FSC_{0.5}$ ↓ | FR↓ | $FSC_{AUC}$ ↑ | $FSC_{0.5}$ ↓ |
|---|---|---|---|---|---|---|---|---|---|
| After degradation | – | 0.6623 | 4.15 | – | 0.6324 | 4.27 | – | 0.6111 | 4.38 |
| DeepEMhancer | 0.22 | $0.80 \pm 0.03$ | $3.2 \pm 0.1$ | 0.22 | $0.79 \pm 0.05$ | $3.3 \pm 0.2$ | 0.22 | $0.77 \pm 0.05$ | $\mathbf{3.4 \pm 0.2}$ |
| spIsoNet | **0.00** | $0.65 \pm 0.01$ | $4.15 \pm 0.01$ | **0.00** | $0.62 \pm 0.01$ | $4.27 \pm 0.03$ | **0.00** | $0.61 \pm 0.01$ | $4.37 \pm 0.03$ |
| CRYOFM | **0.00** | $\mathbf{0.88 \pm 0.03}$ | $\mathbf{3.1 \pm 0.1}$ | **0.00** | $\mathbf{0.84 \pm 0.03}$ | $\mathbf{3.2 \pm 0.1}$ | **0.00** | $\mathbf{0.81 \pm 0.04}$ | $\mathbf{3.4 \pm 0.2}$ |

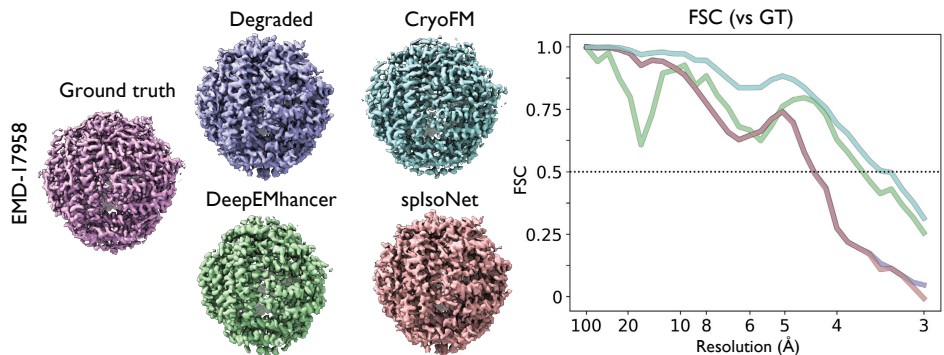

Figure 5: Result of the anisotropic noise denoising task. Two density maps from EMDB are added anisotropic spectral noise that the estimated resolution is 4.38 Å. The degraded density maps are filtered by *relion_postprocess* for visual clarity.

methods like spIsoNet lack a data-driven foundation, limiting their effectiveness. Additionally, spIsoNet requires over 5 hours on a V100 GPU for one dataset, whereas CRYOFM completes the task in about 30 minutes.

### 5.3 MISSING WEDGE RESTORATION

The missing wedge problem in cryo-ET arises from the limited tilt range of the electron microscope during data acquisition, causing incomplete Fourier space sampling and artifacts in the subtomograms. We simulate this effect by applying a wedge-shaped mask in the Fourier domain, removing data from unmeasured orientations. The forward model $\mathcal{A} : \mathbb{C}^{D \times D \times D} \to \mathbb{C}^{D \times D \times D}$ is given by:

$$\mathcal{A}(\tilde{\mathbf{V}}) = \begin{cases} \tilde{\mathbf{V}}, & \text{if } \theta_{\min} \leq \theta \leq \theta_{\max} \\ 0, & \text{otherwise} \end{cases},$$

where $\theta_{\min}$ and $\theta_{\max}$ are typically set to $-60°$ and $+60°$ respectively, representing the common tilt angle limits in experimental setups as illustrated in Fig. 4. For posterior sampling, we slightly modify the flow posterior sampling algorithm to get Alg. 2. The modification is based on the fact that it is not necessary to compute the gradient of the loss function, since we can directly combine the observed part in $\mathbf{y}$ with the remaining part in $\mathbf{x}_t$ to maximize the likelihood.

We apply the missing wedge effect to the test set, resulting in Tab. 3 and an example in Fig. 6. CRYOFM effectively restores the original signal, reducing the missing wedge artifacts, as shown by both FSC and visual inspection. The FSC curve confirm a strong correlation with the ground truth, particularly in the low-frequency range.

Table 3: Results of the missing wedge restoration task.

| | $FSC_{AUC}$ ↑ | $FSC_{AUC}$ ↑ (Missing Region) |
|---|---|---|
| After degradation | $0.80 \pm 0.02$ | $0.0000$ |
| CRYOFM | $0.92 \pm 0.02$ | $0.76 \pm 0.06$ |

### 5.4 *Ab initio* MODELING

*Ab initio* modeling in cryo-EM involves reconstructing a coarse density map from 2D particle projections, which serves as a reference for refinement. Here, we simplify the problem by focusing on generating a coarse density map from a few clean 2D projections obtained via upstream 2D classification. This approach mirrors earlier *ab initio* methods (Ludtke et al., 1999; Voss et al., 2010) before cryoSPARC's introduction of SGD (Punjani et al., 2017). Given $K$ projections, there exist

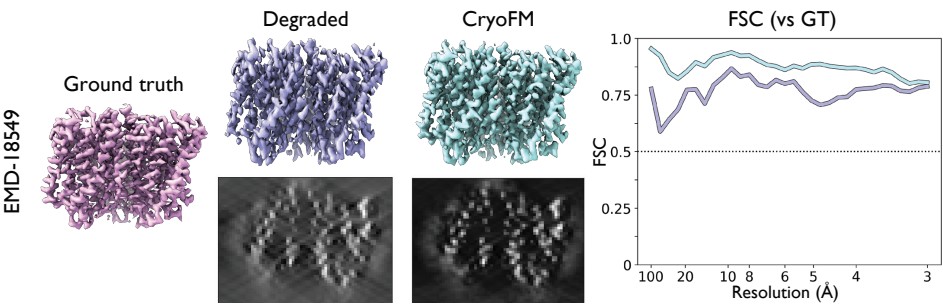

Figure 6: Result of the missing wedge restoration task. Two maps from EMDB are masked in Fourier space to simulate the missing wedge effect in cryo-ET. The central slices through the maps are also shown.

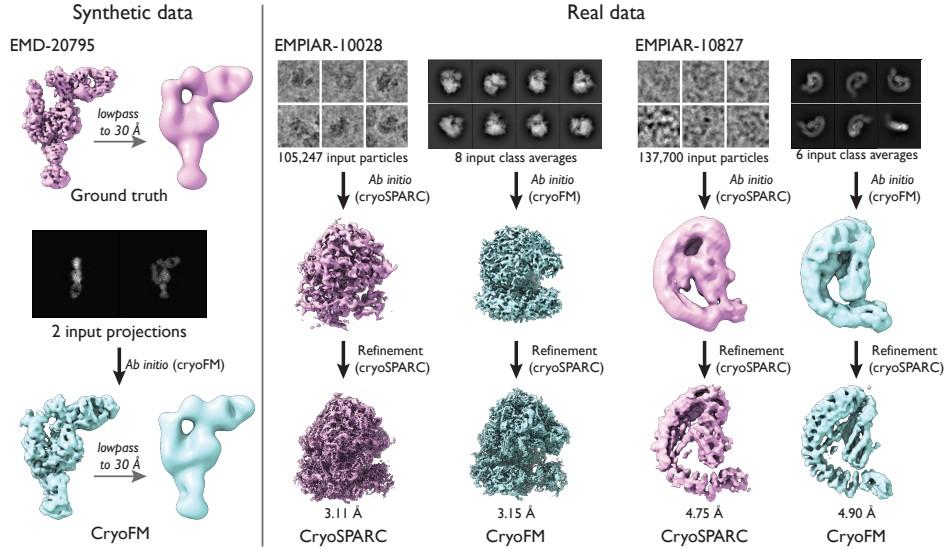

Figure 7: Result of the *ab initio* modeling task. One synthetic data and two real data are shown.

$K$ forward operators, where the $k$-th forward model is $\mathcal{A}^{(k)} : \mathbb{R}^{D \times D \times D} \to \mathbb{R}^{D \times D}$ given by:

$$\mathcal{A}^{(k)}(\boldsymbol{V}) = \mathcal{P}(\phi^{(k)}, \boldsymbol{V}), \quad k \in [1, 2, \cdots, K],$$

where $\mathcal{P}$ is a projection operator in the real space, and $\phi^{(k)} \in \mathbb{SO}(3) \times \mathbb{R}^2$ is the pose of the 2D projections. In this task, $\phi^{(k)}$ is unknown and can not be pre-determined. We modify the flow posterior sampling to get Alg. 3, where the main modifications are: (i) searching the optimal pose iteratively in the sampling process (Scheres, 2012a; Punjani et al., 2017; Zhong et al., 2021b), and (ii) computing the likelihood using correlation to avoid numerical issues. Moreover, since *ab initio* modeling focuses on capturing the global shape at low resolution, we use CRYOFM-L for this task.

We apply CRYOFM to one synthetic dataset and two real datasets from the Electron Microscopy Public Image Archive (EMPIAR). For the synthetic dataset, we use 2 clean projections, while for the real datasets, we use 8 or 6 selected class averages as the input. As shown in Fig. 7, CRYOFM closely matches the ground truth at low frequencies on the synthetic dataset. For the real datasets, CRYOFM produces *ab initio* models that achieve similar final resolutions after refinement as those generated by cryoSPARC, demonstrating its potential for 3D reconstruction in cryo-EM.

## 5.5 ABLATION & DISCUSSION

In this section, we perform ablation studies to analyze the impact of different design choices and hyper-parameters. We briefly present some conclusions here and refer the reader to Appendix H.

**Moderate patchifying and downsampling parameters lead to efficient training with minimal performance degradation.** As shown in Fig. 8, reducing both patchifying and downsampling parameters lowers the test loss, since less downsampling causes less information loss. However, the

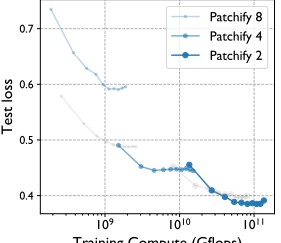 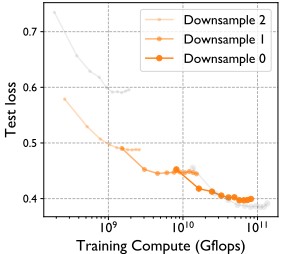 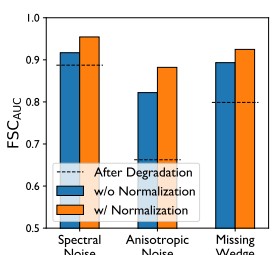

Figure 8: MSE loss on test set as a function of total training compute. The estimation of training compute is consistent with Peebles & Xie (2023).

Figure 9: Downstream performance with respect to data normalization.

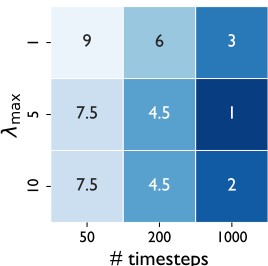 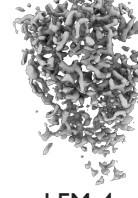 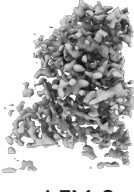 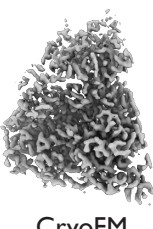

LFM-4          LFM-8          CryoFM

Figure 10: Averaged ranking ($\downarrow$) across downstream tasks for hyperparameters.

Figure 11: Unconditional sampling from models in different spaces. For latent flow models (LFM), the high-frequency information contains more noise.

reduction becomes marginal while the training compute (Gflops) increases significantly. To achieve a balance between the performance and training cost, we set the patchifying parameter to 4 and the downsampling parameter to 1, respectively.

**Data normalization enhances downstream performance across all tasks.** Data normalization is the last step in data processing (as referenced in Section C.2). A density with a value ranging from $[0, 1]$ is subtracted by $0.04$ and then divided by $0.09$. Fig. 9 demonstrates that models trained with normalized data consistently improves the $\text{FSC}_{\text{AUC}}$ metrics compared to the one without normalization in three downstream tasks. The observation is in line with some DDPM-based work (Yim et al., 2023b; Rombach et al., 2022), as a significant change in the variance of the data can make the model difficult to train (Karras et al., 2022).

**More sampling steps boosts the performance in density restoration.** Fig. 10 presents the averaged ranking across three downstream tasks for different time steps (#timesteps) and the maximum step sizes ($\lambda_{\max}$). The main factor influencing performance is the number of time steps. A smaller $\lambda_{\max}$ may reduce the impact of the likelihood term and thus degrade performance. We choose the best combination: $\lambda_{\max} = 5$ and $\#\text{timesteps} = 1000$.

**Flow models trained in the voxel space converge much faster and better than the latent flow models.** We conducted experiments on training the model in the latent space (Rombach et al., 2022). Fig. 11 illustrates the unconditional sampling results from two latent flow models (LFM), where the high-frequency information is more noisy. The observation concides with that of Crowson et al. (2024), where they found that latent diffusion models fail to generate fine details. We suspect that the VAE is under-trained due to a small amount of data, and we will leave this for future work.

## 6 CONCLUSION

In this study, we present CRYOFM, a flow matching-based foundation model that learns the *prior* distribution of high-quality cryo-EM densities. During inference, we derive the forward operators for specific tasks, allowing protein densities to be sampled from the posterior distribution based on given observations. CRYOFM demonstrates versatility by restoring protein densities across four distinct tasks without fine-tuning, showcasing the potential of deep generative models as a powerful prior in cryo-EM. While promising, CRYOFM has limitations: though we use real data with synthetic noise for comparison, applying the method directly to real-world noisy densities remains challenging. Additionally, our work does not address reconstructing 3D densities from raw 2D particles, but we believe CRYOFM can contribute to solving these complex tasks in future work.

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

## A DISCUSSION ON FLOW MATCHING

### A.1 CONCEPTS IN FLOW MATCHING

Fig. 12 illustrates three main concepts in flow matching. Here, the horizontal bar represents the time dimension, and each vertical slice represents a time-dependent probability distribution with only one dimension.

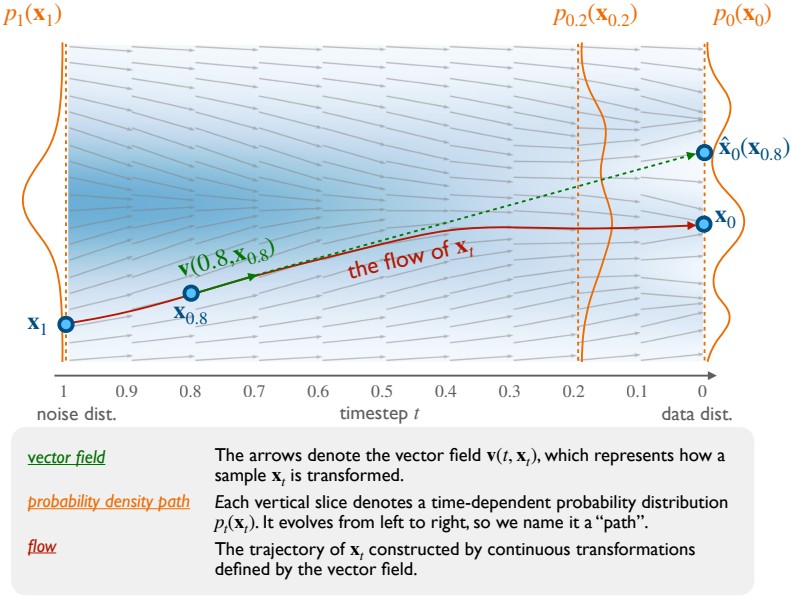

Figure 12: An illustration of three main concepts in flow matching: (i) the vector field $\mathbf{v}_\Theta(t, \mathbf{x}_t)$, (ii) the probability density path $p_t(\mathbf{x}_t)$, and (iii) the flow of $\mathbf{x}_t$. The approximated posterior mean $\hat{\mathbf{x}}_0(\mathbf{x}_t)$ in Alg. 1 is also depicted on the right part.

### A.2 CHOICE OF FLOW MATCHING FOR FOUNDATION MODELS

In this section, we discuss the choice of employing flow matching as the backbone of the cryo-EM foundation model. The main motivation is that (i) generative modeling is a trend in building modern foundation models, and (ii) flow matching excels at generative modeling.

**Generative modeling is a trend in building modern foundation models.** ChatGPT (Achiam et al., 2023) has disrupted the entire NLP (natural language processing) area with its extremely strong generative power. It brings about a paradigm shift from *representation learning* to *generative modeling*. For representation learning, one of the most prominent models is BERT (Devlin, 2018), which performs well in language understanding but is not adept at performing generation. Similar trends

can also be observed in computational biology. Two typical cases in this field are ESM-3 and AlphaFold3. Built upon ESM (Rives et al., 2021) (a BERT-style model), Hayes et al. (2024) developed ESM-3 that has the ability to generate protein sequences. A major change from AlphaFold2 (Jumper et al., 2021) to AlphaFold3 (Abramson et al., 2024) is the diffusion module that can be utilized to generate an ensemble of structures.

**Flow matching excels at generative modeling.** Diffusion-based foundation models have had a significant impact on generative modeling in AI for biology. AlphaFold3 (Abramson et al., 2024), RFDiffusion (Watson et al., 2023), and Chroma (Ingraham et al., 2023) are all based on diffusion models (DDPM). ESM-3 (Hayes et al., 2024), although not explicitly stated in its manuscript, is closely related to discrete diffusion models. One of the crucial reasons for the success of DDPM resides in iterative refinement, which gradually refines the generated samples to a more superior state. The iterative nature of DDPM enables it to outperform its counterparts with one-step generation methods, such as VAE and GAN. Flow matching can be regarded as a variant of DDPM and shares many similar properties with DDPM. We prefer flow matching over DDPM since it is easier to train and can converge faster. In the early stage of this study, we trained both DDPM and flow matching. We found that flow matching can yield similar results while incurring less computational cost.

# B  ADDITIOIANT ONAL RELATED WORK

## B.1  DENSITY MODIFICATION AND DENOISING IN CRYO-EM

Traditional density modification methods often apply Fourier space weighting, utilizing frequency-dependent scaling to suppress noise and enhance signal (Jakobi et al., 2017; Ramírez-Aportela et al., 2020; Terwilliger et al., 2020a; Kaur et al., 2021). These heuristic-based approaches, like Wiener-style deblurring (Ramírez-Aportela et al., 2020), tend to rely on local resolution estimates and filtering strategies but may struggle with intricate structural details due to limited priors. On the deep learning side, Blush (Kimanius et al., 2024) further refine density maps by denoising half-maps during iterative refinement. However, they can introduce hallucinated details in lower-resolution maps. In contrast, methods such as M (Tegunov et al., 2021) and spIsoNet (Liu et al., 2024) avoid external data, with M leveraging noise2noise (Moran et al., 2020) for denoising independent half-maps, and spIsoNet addressing anisotropic signal distributions through self-supervised learning. Though slower, these approaches tend to reduce the risk of hallucinations and offer better robustness, but their power is often limited by not being data-driven. In this paper, we selected DeepEMhancer (Sanchez-Garcia et al., 2021), EMReady (He et al., 2023) and spIsoNet as the baselines for the denoising tasks.

**DeepEMhancer** DeepEMhancer is a deep learning-based tool designed to enhance cryo-EM density maps, improving their interpretability and aiding in structural analysis. The method employs a convolutional neural network (CNN) trained on pairs of experimental cryo-EM maps and corresponding locally sharpened maps. This training enables DeepEMhancer to learn the mapping between low-quality input maps and their high-quality counterparts. The pretrained model processes new cryo-EM maps to produce enhanced versions with improved clarity and detail, facilitating more accurate structural interpretations.

**EMReady** Similar to DeepEMhancer, EMReady is a computational method designed to enhance cryo-EM density maps by simultaneously applying local and non-local denoising techniques. The method uses a Swin-Conv-3DUNet and trained on pairs of experimental cryo-EM maps and corresponding synthetic maps from the atomic models. The pretrained model can enhance cryo-EM maps for better interpretations.

**spIsoNet** spIsoNet is a self-supervised deep learning method developed to address the preferred orientation problem in single-particle cryo-EM. This issue arises when particles predominantly adopt specific orientations during imaging, leading to anisotropic data and potential inaccuracies in 3D reconstructions. The method adopts a U-Net architecture and leverages the half maps in cryo-EM reconstruction to learn representations from the overrepresented orientations, recovering information for the underrepresented or missing views.

## B.2  MISSING WEDGE PROBLEM IN CRYO-ET

Cryo-ET enables detailed visualization of macromolecular complexes and cellular structures in their native environments. The missing wedge problem occurs due to the physical limitation of tilt angles

during data acquisition, typically restricted to $(-60°, +60°)$, resulting in missing data in Fourier space and artifacts such as elongation along the missing axes. Traditional methods that use signal processing techniques and regularization strategies (Goris et al., 2012; Deng et al., 2016; Yan et al., 2019; Zhai et al., 2020) are often based on heuristic assumptions and have limitations in fully recovering lost data. Recently, deep learning models have been applied to this problem, offering improved recovery of complex patterns in tomograms (Liu et al., 2022b; Van Veen et al., 2024). However, these methods often focus on entire tomograms, making it difficult to incorporate specific prior knowledge of protein structures, limiting their effectiveness in subtomogram reconstructions.

### B.3 *Ab initio* MODELING IN CRYO-EM

Earlier approaches to *ab initio* modeling relied on experimental techniques, such as image tilt pairs (Radermacher et al., 1986; Leschziner & Nogales, 2006), which provided indirect pose information, or negative stain (De Carlo & Harris, 2011), which improved SNR but at the cost of high-frequency detail. Computationally, 2D class averages were commonly used as input due to their higher SNR compared to raw particles, though they had limitations, particularly in fully sampling Fourier space due to the restricted range of particle orientations (Voss et al., 2010). While cryoSPARC's SGD approach improved this by working directly with raw particles (Punjani et al., 2017), certain structural features or heterogeneity observed in 2D class averages may still be lost in the final 3D reconstruction, especially in challenging samples.

### B.4 DIFFUSION/FLOW-BASED MODEL

Flow-based models (Lipman et al., 2022; Liu et al., 2022a) and denoising diffusion probablistic models (DDPM) (Sohl-Dickstein et al., 2015; Ho et al., 2020) are two modern deep generative models. They exhibit many similarities in technical details. DDPM implements an iterative refinement process by learning to gradually denoise a sample from a normal distribution. It has achieved the state-of-the-art results on many generative tasks, including image generation (Rombach et al., 2022; Podell et al., 2023; Peebles & Xie, 2023), video generation (Blattmann et al., 2023), and molecule generation (Yim et al., 2023b; Abramson et al., 2024; Ingraham et al., 2023; Watson et al., 2023; Wang et al., 2024b), etc. Recently, diffusion models have been adopted in the cryo-EM field. Kreis et al. (2022) traverses the latent space of cryoDRGN (Zhong et al., 2021a) with a diffusion model, while Wang et al. (2024a) refines structures for model building by iteratively denoising the density. Flow-based models regress a vector field that generate a disired probability path. Their simple and efficient implementation enables fast learning. These models have shown success in various domains, including image generation (Esser et al., 2024) and moleculer generation (Yim et al., 2023b; Bose et al., 2023). The DDPM objective can also be unified into flow-based models by converting it into a probability flow ODE (Song et al., 2021b).

### B.5 VISION TRANSFORMERS FOR DIFFUSION MODELS

Diffusion transformers (Peebles & Xie, 2023) have demonstrated significant scalability and generative capabilities in image-related tasks (Esser et al., 2024; Hoogeboom et al., 2023; Hatamizadeh et al., 2024; Zhou et al., 2024). In particular, HDiT (Crowson et al., 2024) leverages the inherent hierarchical nature of visual patterns in its model design. By integrating the characteristics of Diffusion Transformer (Peebles & Xie, 2023) and Hourglass transformers (Nawrot et al., 2022), and employing local attention mechanisms (Hassani et al., 2023), HDiT offers an efficient model structure suitable for training the diffusion model in the data space.

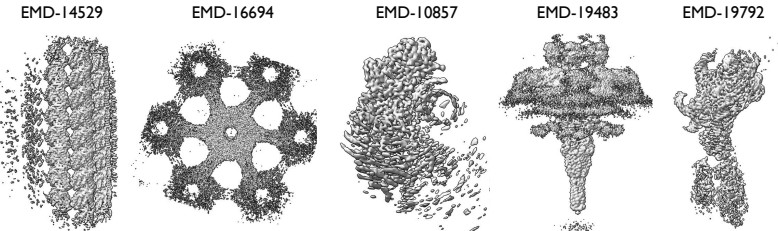

Figure 13: Examples of problematic cases removed from the training dataset after manual curation. Low contour levels are used to highlight the heterogeneous local resolution.

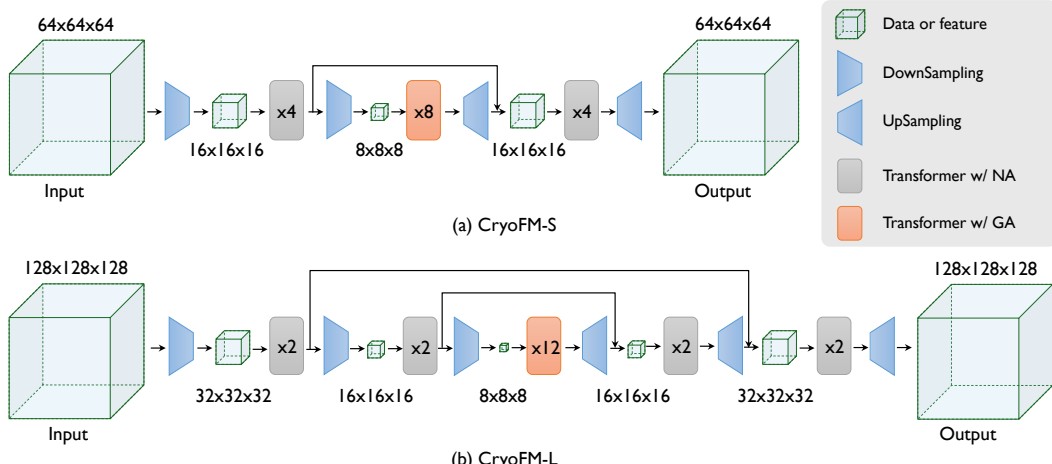

Figure 14: Overview of two 3D CRYOFM model details. (a) CRYOFM-S design for input shape $64^3$ and (b) CRYOFM-L for shape $128^3$.

## C    ADDITIONAL DATA CURATION AND PROCESSING

### C.1    DATA CURATION

Examples of the problematic cases which were excluded from the training set after manual curation are shown in Fig. 13. These includes density maps with very heterogeneous local resolutions that significant portions of the maps are poorly resolved. The EMDB IDs of the training and testing data used in this paper have been uploaded to https://figshare.com/s/9ef2614108391c04d910.

### C.2    DATA STANDARDIZATION

We preprocessed the deposited density maps to normalize their values to a consistent range of $[0, 1]$. Since the absolute values of density maps are not intrinsically meaningful and the value ranges vary significantly, we applied a uniform processing procedure as follows: **Clipping**: We clipped the values of each density map based on the minimum value, which is set to 0, and the 99.999th percentile value of the density map. **Scaling**: Subsequently, the clipped data were scaled to the $[0, 1]$ range by adjusting according to the new minimum and maximum values obtained after clipping. **Normalization**: we used a mean of $0.04$ and a standard deviation of $0.09$, which were calculated based on the data that have been scaled to the $[0, 1]$ range.

## D    ADDITIONAL DETAILED INFORMATION OF CRYOFM

### D.1    IMPLEMENTATION DETAILS

Fig. 14 illustrate the model architectures for input dimensions of $64^3$ and $128^3$. Fig. 15 shows the details of different attention structure we used in the transformer block. Tab. 4 details the specific model parameter configurations (aligned with the naming conventions used in Crowson et al. (2024)) and the training hyperparameters.

In all experiments, we employed the FairseqAdam (Ott et al., 2019) optimizer with a default learning rate of 1e-4, betas set to (0.9, 0.98), and a weight decay of 0.01. A linear warm-up strategy was applied during the first 2000 steps of training.

### D.2    LIKELIHOOD ESTIMATION

Given a data point $\mathbf{x}_0 \in \mathbb{R}^n$, we estimate its likelihood by solving a probability flow ODE (Song et al., 2021b; Lipman et al., 2022; Chen et al., 2018). We start with the continuity equation of a velocity field $\mathbf{v}_\Theta : [0, 1] \times \mathbb{R}^n \to \mathbb{R}^n$:

$$\frac{d}{dt} \log p_t(\mathbf{x}_t) + \nabla \cdot \mathbf{v}_\Theta(t, \mathbf{x}_t) = 0.$$

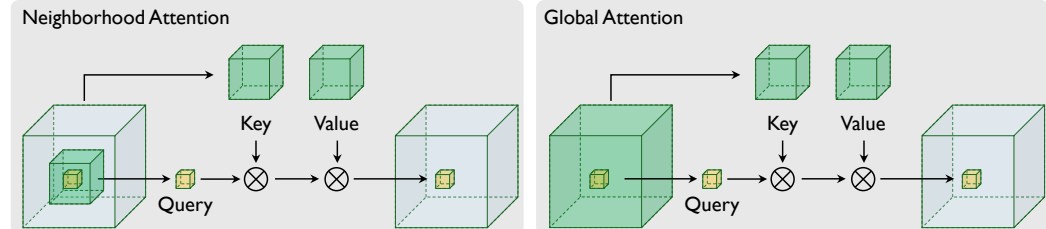

Figure 15: Illustration of QKV structure of neighborhood attention (NA) and global attention (GA).

Table 4: Details of training and model setup.

| Parameter | CRYOFM-S | CRYOFM-L |
|---|---|---|
| Parameters | 335.18 M | 308.54 M |
| GFLOP/forward | 395.87 | 427.26 |
| Training Steps | 150k | 300k |
| Batch Size | 128 | 128 |
| Precision | bf16 | bf16 |
| Training Hardware | 8×A100 | 8×A100 |
| Patchifying | 4 | 4 |
| Levels (Local + Global Attention) | 1 + 1 | 2 + 1 |
| Depth | [4, 8] | [2, 2, 12] |
| Widths | [768, 1536] | [320, 640, 1280] |
| Attention Heads (Width / Head Dim) | [12, 24] | [5, 10, 20] |
| Attention Head Dim | 64 | 64 |
| Neighborhood Kernel Size | 7 | 7 |

Integrating $t \in [0, 1]$ gives:

$$\log p_0(\mathbf{x}_0) = \log p_1(\mathbf{x}_1) - \int_0^1 \nabla \cdot \mathbf{v}_\Theta(t, \mathbf{x}_t) dt.$$

Since $p_1$ is a standard normal distribution, $\log p_1(\mathbf{x}_1)$ can be calculated exactly. The second term in the right-hand side of the equation can be computed by solving an ODE:

$$\frac{d}{dt} \begin{bmatrix} \mathbf{x}_t \\ f_t \end{bmatrix} = \begin{bmatrix} \mathbf{v}_\Theta(t, \mathbf{x}_t) \\ \nabla \cdot \mathbf{v}_\Theta(t, \mathbf{x}_t) \end{bmatrix},$$

with initial conditions $\mathbf{x}_0$ and $f_0 = 0$. Since the divergence operator requires a lot of computational cost, we follow Grathwohl et al. (2018) to use the Hutchinson trace estimator to get an unbiased estimate:

$$\frac{d}{dt} \begin{bmatrix} \mathbf{x}_t \\ f_t \end{bmatrix} = \begin{bmatrix} \mathbf{v}_\Theta(t, \mathbf{x}_t) \\ \mathbf{z}^\top \nabla \mathbf{v}_\Theta(t, \mathbf{x}_t) \mathbf{z} \end{bmatrix},$$

where $\mathbf{z} \sim \mathcal{N}(0, \mathbf{I})$. In practice, we use the Runge–Kutta method (Dormand & Prince, 1980) to solve the ODE to get $f_1$, and the log probability is $\log p_0(\mathbf{x}_0) = \log p_1(\mathbf{x}_1) - f_1$.

**Normalization** A commonly employed trick for training flow models is to linearly transform a data point, so that the transformed data are centered around $0$ and the standard deviation is not too off. An additional step is required to estimate the likelihood of the original data point, since the transformation changes the probability density. Given a volume density in the dataset $\mathbf{x} \in \mathbb{R}^{64 \times 64 \times 64}$, if we normalize it by $\mathbf{y} = (\mathbf{x} - 0.04)/0.09$ and compute the log-probability of $\log p(\mathbf{y})$, then the probability of $\mathbf{x}$ can be further calculated by:

$$\log p(\mathbf{x}) = \log p(\mathbf{y}) + \log \left( \frac{1}{0.09} \right)^{64^3}$$

$$= \log p(\mathbf{y}) + 64^3 \log \left( \frac{1}{0.09} \right)$$

$$\approx \log p(\mathbf{y}) + 631,228$$

Table 5: Additional results of the spectral noise denoising task, comparing CRYOFM with Deep-EMhancer (Sanchez-Garcia et al., 2021) and EMReady (He et al., 2023).

| | Estimated resolution at different level of noise | | | | | |
| | 8.5 Å | | | 15.0 Å | | |
| | FR$\downarrow$ | FSC$_{AUC}$ $\uparrow$ | FSC$_{0.5}$ $\downarrow$ | FR$\downarrow$ | FSC$_{AUC}$ $\uparrow$ | FSC$_{0.5}$ $\downarrow$ |
|---|---|---|---|---|---|---|
| After degradation | - | 0.3275 | 8.60 | - | 0.2014 | 15.98 |
| DeepEMhancer | 0.13 | $0.37 \pm 0.02$ | $\textbf{7.4} \pm \textbf{0.3}$ | 0.03 | $0.16 \pm 0.04$ | $17.5 \pm 2.5$ |
| EMReady | $\textbf{0.00}$ | $0.37 \pm 0.04$ | $7.7 \pm 0.8$ | 0.03 | $0.22 \pm 0.02$ | $\textbf{13.3} \pm \textbf{1.6}$ |
| CRYOFM | $\textbf{0.00}$ | $\textbf{0.38} \pm \textbf{0.01}$ | $7.5 \pm 0.3$ | $\textbf{0.00}$ | $\textbf{0.24} \pm \textbf{0.01}$ | $15.0 \pm 0.7$ |

## E  ADDITIONAL INFORMATION OF THE EXPERIMENTS

### E.1  METRICS

**FSC$_{AUC}$**  The area under the curve (AUC) of the FSC curve (FSC$_{AUC}$) provides an overall correlation across all spatial frequencies, offering a comprehensive view of the alignment between the ground truth and the restored density map (Harauz & van Heel, 1986).

**FSC$_{0.5}$**  The FSC resolution at the 0.5 cutoff (FSC$_{0.5}$) is a standard metric in cryo-EM, indicating the resolution of the reconstructed map relative to the ground truth (Rosenthal & Henderson, 2003).

**Fail Rate (FR)**  The fail rate is defined as the portion of the test cases where the method either fails to run or produces an FSC$_{0.5}$ between the result and the ground truth greater than 30 Å. This metric helps identify cases where the reconstruction process fails to achieve meaningful results, ensuring a more comprehensive evaluation of the method's robustness.

Both FSC$_{AUC}$ and FSC$_{0.5}$ are reported as the mean and standard deviation across the 32 density maps in the test set, excluding failed cases.

### E.2  SPECTRAL NOISE DENOISING

The results of spectral noise denoising at additional resolutions are presented in Tab. 5.

## F  ESTIMATION OF THE FORWARD OPERATORS

### F.1  SPECTRAL NOISE POWER ESTIMATION

Given two half maps from a cryo-EM reconstruction, $\mathbf{y}_1$ and $\mathbf{y}_2$ (observations), where the signal and noise power are the same and the signal and noise are uncorrelated, the obeservation models for $\mathbf{y}_1$ and $\mathbf{y}_2$ are:

$$\tilde{\mathbf{y}}_1 = \tilde{\mathbf{V}} + \epsilon_1, \quad \epsilon_1(\nu) \sim \mathcal{N}(0, \sigma^2_{\text{noise}}(\nu)),$$

$$\tilde{\mathbf{y}}_2 = \tilde{\mathbf{V}} + \epsilon_2, \quad \epsilon_2(\nu) \sim \mathcal{N}(0, \sigma^2_{\text{noise}}(\nu)).$$

The so-called "Gold-Standard" Fourier Shell Correlation (GSFSC) is the FSC between two half maps, which is defined as:

$$\text{GSFSC}(\nu) \triangleq \frac{\tilde{\mathbf{y}}_1(\nu) \cdot \tilde{\mathbf{y}}_2(\nu)}{\|\tilde{\mathbf{y}}_1(\nu)\| \cdot \|\tilde{\mathbf{y}}_2(\nu)\|},$$

where $\tilde{\mathbf{y}}(\nu)$ represents the component of $\tilde{\mathbf{y}}$ at frequency $\nu$ (i.e., all values on a spherical shell), and $\|\tilde{\mathbf{y}}(\nu)\|^2$ is the radial power spectrum.

From the GSFSC, the signal-to-noise ratio (SNR) can be computed as (Rosenthal & Henderson, 2003):

$$\text{SNR}(\nu) = \frac{\mathbb{E}[\sigma^2_{\text{signal}}(\nu)]}{\mathbb{E}[\sigma^2_{\text{noise}}(\nu)]} = \frac{\text{GSFSC}(\nu)}{1 - \text{GSFSC}(\nu)}.$$

Since $\mathbf{y}_1$ and $\mathbf{y}_2$ share the same signal and noise at each frequency, and the signal $\tilde{\mathbf{V}}$ and the noise $\epsilon$ are uncorrelated, we have:

$$\mathbb{E}[\|\tilde{\mathbf{y}}(\nu)\|^2] = \mathbb{E}[\|\tilde{\mathbf{V}}(\nu) + \epsilon(\nu)\|^2] = \mathbb{E}[\sigma_{\text{signal}}^2(\nu)] + \mathbb{E}[\sigma_{\text{noise}}^2(\nu)].$$

Thus,

$$\mathbb{E}[\|\tilde{\mathbf{y}}_1(\nu)\|^2] = \mathbb{E}[\|\tilde{\mathbf{y}}_2(\nu)\|^2] = \mathbb{E}[\sigma_{\text{signal}}^2(\nu)] + \mathbb{E}[\sigma_{\text{noise}}^2(\nu)].$$

Given the known SNR, we can derive:

$$\mathbb{E}[\sigma_{\text{noise}}^2(\nu)] = \frac{\mathbb{E}[\|\tilde{\mathbf{y}}_1(\nu)\|^2]}{1 + \text{SNR}(\nu)} = \frac{\mathbb{E}[\|\tilde{\mathbf{y}}_2(\nu)\|^2]}{1 + \text{SNR}(\nu)},$$

and

$$\mathbb{E}[\sigma_{\text{signal}}^2(\nu)] = \frac{\mathbb{E}[\|\tilde{\mathbf{y}}_1(\nu)\|^2] \cdot \text{SNR}(\nu)}{1 + \text{SNR}(\nu)} = \frac{\mathbb{E}[\|\tilde{\mathbf{y}}_2(\nu)\|^2] \cdot \text{SNR}(\nu)}{1 + \text{SNR}(\nu)}.$$

### F.2 Anisotropic noise power estimation

Anisotropic noise means that the noise is not uniform across all directions; in certain directions, the noise power is stronger or weaker than in others. In the context of spectral noise, anisotropic noise implies that the variance of the noise depends on the direction in Fourier space. We denote by $\xi = (\phi, \theta)$ the anisotropic angular components on the sphere, and by $\nu$ the frequency component, corresponding to the radial component in spherical coordinates. We assume that the signal power $\sigma_{\text{signal}}^2(\nu)$ is isotropic, meaning $\sigma_{\text{signal}}^2(\xi, \nu)$ is constant for different values of $\xi$.

The noise power averaged over all directions for a given frequency $\nu$ is denoted as:

$$h(\nu) = \frac{1}{N_\nu} \sum_\xi \sigma_{\text{noise}}^2(\xi, \nu),$$

where $N_\nu$ represents the number of voxels in the frequency shell at $\nu$.

Our goal is to estimate the anisotropic noise power, $\sigma_{\text{noise}}^2(\xi, \nu)$, in spherical coordinates. We assume that for a given angle $\xi$, the noise power $\sigma_{\text{noise}}^2(\xi, \nu)$ can be expressed as:

$$\sigma_{\text{noise}}^2(\xi, \nu) = \alpha(\xi, \nu) \cdot \sigma_{\text{noise}}^2(\xi_0, \nu),$$

where $\xi_0$ represents an arbitrary reference direction in spherical coordinates, serving as the baseline for the noise power $\sigma_{\text{noise}}^2(\xi, \nu)$, with $\alpha(\xi, \nu)$ adjusting the noise power in other directions. In practice, $\alpha(\xi, \nu)$ can be inferred from the number of particles contributing to each pose during the reconstruction process.

Substituting the anisotropic noise power expression, we can derive:

$$h(\nu) = \frac{1}{N_\nu} \sum_\xi \alpha(\xi, \nu) \cdot \sigma_{\text{noise}}^2(\xi_0, \nu).$$

Simplifying, we obtain:

$$h(\nu) = \frac{\sigma_{\text{noise}}^2(\xi_0, \nu)}{N_\nu} \sum_\xi \alpha(\xi, \nu),$$

where $h(\nu)$ is the average noise power derived from the the half-map GSFSC in Section F.1.

Thus, we can solve for $\sigma_{\text{noise}}^2(\xi_0, \nu)$ as:

$$\sigma_{\text{noise}}^2(\xi_0, \nu) = \frac{h(\nu) \cdot N_\nu}{\sum_\xi \alpha(\xi, \nu)}.$$

Finally, the anisotropic noise power for any angle $\xi$ is given by:

$$\sigma_{\text{noise}}^2(\xi, \nu) = \alpha(\xi, \nu) \cdot \frac{h(\nu) \cdot N_\nu}{\sum_\xi \alpha(\xi, \nu)}.$$

---

**Algorithm 2** Flow Posterior Sampling for the Missing Wedge Problem

---

**Require:** a pretrained vector field $\mathbf{v}_\Theta : [0, 1] \times \mathbb{R}^n \to \mathbb{R}^n$, a forward operator $\mathcal{A} : \mathbb{R}^n \to \mathbb{R}^m$, an obseravation $\mathbf{y} \in \mathbb{R}^m$, number of steps $N$
**Ensure:** the recovered signal $\mathbf{x}_0$
    $\Delta t \leftarrow \frac{1}{N}$
    $\mathbf{x}_1 \leftarrow \mathcal{N}(0, \boldsymbol{I})$
    **for** $t \in [1, 1 - \Delta t, 1 - 2\Delta t, \cdots, \Delta t]$ **do**
        $\hat{\mathbf{x}}_0 \leftarrow \mathbf{x}_t - t \cdot \mathbf{v}_\Theta(t, \mathbf{x}_t)$
        $\hat{\mathbf{x}}_0' \leftarrow \mathcal{A}^\dagger \mathbf{y} + (1 - \mathcal{A}^\dagger)\hat{\mathbf{x}}_0$                      $\triangleright$ Fill the observed part $\mathbf{y}$ into $\hat{\mathbf{x}}_0$
        $\mathbf{x}_{t-\Delta t} \leftarrow \frac{t-\Delta t}{t} \cdot \mathbf{x}_t + \frac{\Delta t}{t} \cdot \hat{\mathbf{x}}_0'$
    **end for**
    **return** $\mathbf{x}_0$

---

## G    Additional Algorithms

In this section, we present algorithms for addressing different downstream tasks within the flow posterior sampling framework. For spectral noise and anisotropic noise denoising, it is natural to apply Alg. 1 in the main text since the loss function is well-defined. However, regarding missing wedge restoration and *ab initio* modeling, we can make slight modifications to the algorithm to better solve these problems.

### G.1   Missing Wedge Restoration

For certain tasks, such as missing wedge restoration, it is not necessary to compute the gradient with respect to the loss function between $\mathbf{y}$ and $\mathcal{A}\mathbf{x}$, since we can directly attain the optimum in an analytical form. The missing wedge restoration is an in-painting task, where a noiseless part of the signal is already observed. $\mathcal{A}$ is an operator extracting a part of the signal, hence the inverse of $\mathcal{A}$ just fills the observed part into the signal. In Alg. 2, we formulate it as the pseudo-inverse operator $\mathcal{A}^\dagger : \mathbb{R}^m \to \mathbb{R}^n$ (Kawar et al., 2021; 2022).

### G.2   *Ab initio* Modeling

For *ab initio* modeling, the $k$-th observed volume (we will name it the *projection* later) is $\mathcal{A}^{(k)}(\boldsymbol{V}) = \mathcal{P}(\phi^{(k)}, \boldsymbol{V})$, where $\phi^{(k)} \in \mathbb{SO}(3) \times \mathbb{R}^2$ is the pose. The posterior sampling of *ab initio* modeling is more complex than the previous tasks in that the pose $\phi^{(k)}$ can not be easily pre-determined. The setting is closely related to blind diffusion posterior sampling (Chung et al., 2023a; Kapon et al., 2024; Laroche et al., 2024; Gan et al., 2024), where the forward operator is unknown. We find $\phi^{(k)}$ in the sampling process, by minimizing the discrepancy between the projection and the observation $\boldsymbol{y}^{(k)}$ using a coarse-to-fine strategy (Zhong et al., 2021b). The discrepancy is defined by the negative correlation between a projection $\mathbf{p}$ and an observation $\mathbf{y}$:

$$\psi_{\text{cor}}(\mathbf{p}, \mathbf{y}) = -\frac{\mathbf{p} \cdot \mathbf{y}}{\|\mathbf{p}\| \cdot \|\mathbf{y}\|}.$$

When computing the likelihood, we further introduce two variants of the loss function to make the sampling process more robust, since this setting is highly ill-posed compared to others. The first one applies a stop gradient operator (denoted by "sg") to the norm of the projection, where the stop gradient operator is identity at forward computation time but has zero partial derivatives (Van Den Oord et al., 2017; Roy et al., 2018; Jang et al., 2016):

$$\psi_{\text{sg}}(\mathbf{p}, \mathbf{y}) = -\frac{\mathbf{p} \cdot \mathbf{y}}{\text{sg}(\|\mathbf{p}\| \cdot \|\mathbf{y}\|)}.$$

This variant does not constrain the norm of the projection, as long as it has a large correlation with the observation. The second one penalizes additional noises in the projection so that it has a value of 0 in the background area of the observation $\mathbf{y}$. The background area is defined by a set of indices $\mathcal{B}(q, \mathbf{y})$ where the corresponding value is smaller the $q$-th percentile of $\mathbf{y}$.

$$\psi_{\text{bg}(q)}(\mathbf{p}, \mathbf{y}) = \sum_{i \in \mathcal{B}(q, \mathbf{y})} \mathbf{p}_i^2.$$

---

**Algorithm 3** Flow Posterior Sampling for *Ab-initio* Modeling

---

**Require:** a pretrained vector field $\mathbf{v}_\Theta : [0, 1] \times \mathbb{R}^n \to \mathbb{R}^n$, $K$ obseravations $\mathbf{y}^{(k)} \in \mathbb{R}^m$, number of steps $N$, maximum step size $\lambda_{\max}$ at each step, the loss function with respect to the likelihood term $\mathcal{L}$

**Ensure:** the recovered signal $\mathbf{x}_0$

    $\Delta t \leftarrow \frac{1}{N}$
    $\mathbf{x}_1 \leftarrow \mathcal{N}(0, \boldsymbol{I})$
    **for** $t \in [1, 1 - \Delta t, 1 - 2\Delta t, \cdots, \Delta t]$ **do**
        $\mathbf{x}'_{t-\Delta t} \leftarrow \mathbf{x}_t - \Delta t \cdot \mathbf{v}_\Theta(t, \mathbf{x}_t)$
        **for** $k \in [1, 2, \cdots, K]$ **do**
            $\phi^{(k)} \leftarrow \underset{\phi \in \mathbb{SO}(3) \times \mathbb{R}^2}{\arg\min} \psi_{\text{cor}}\left(\mathcal{P}(\phi, \hat{\mathbf{x}}_0(\mathbf{x}_t)), \mathbf{y}^{(k)}\right)$         ▷ Search the optimal pose
            $l^{(k)}(\mathbf{x}_t) \leftarrow \mathcal{L}\left(\mathcal{P}(\phi^{(k)}, \hat{\mathbf{x}}_0(\mathbf{x}_t)), \mathbf{y}^{(k)}\right)$         ▷ Loss w.r.t the optimal pose
        **end for**
        $l(\mathbf{x}_t) \leftarrow \frac{1}{K} \sum_k l^{(k)}(\mathbf{x}_t)$
        $\mathbf{g} \leftarrow \frac{\nabla_{\mathbf{x}_t} l(\mathbf{x}_t)}{\|\nabla_{\mathbf{x}_t} l(\mathbf{x}_t)\|_2}$
        $\lambda'_t \leftarrow \min\left\{\lambda_{\max}, \frac{t}{1-t}\right\}$
        $\mathbf{x}_{t-\Delta t} \leftarrow \mathbf{x}'_{t-\Delta t} - \lambda'_t \mathbf{g}$
    **end for**
    **return** $\mathbf{x}_0$

---

The final loss function $\mathcal{L}$ can be a combination of $\psi_{\text{cor}}$, $\psi_{\text{sg}}$ and $\psi_{\text{bg}(q)}$. We tune the parameter on different datasets (Tab. 6) by examining the human preference for the reconstruction results. A full alorithm can be found in Alg. 3.

Table 6: Hyperparameters for *ab initio* modeling

| EMPIAR-ID | stop gradient | $q$ |
|---|---|---|
| 10345 (synthetic) | no | 0.7 |
| 10028 | yes | 0.0 |
| 10827 | no | 0.7 |

## H   ADDITIONAL ABLATION STUDY

### H.1   DATA AUGMENTATION

Protein density data are three-dimensional data characterized by large size differences and lack of canonical orientation. To train the model efficiently, we need to crop the data to the same size. Additionally, to expose the model to enough data with various orientations, we also need to apply appropriate rotational augmentation to the data. In this section, we discuss different data augmentation strategies and how they affect the training and the performance on the downstream tasks. We implemente four different augmentation strategies, as shown in Fig. 16: **center crop**: Only crops a patch from the center of the original data for training; **random crop**: Randomly crops patches from the original data; **rot24**: Achieves data rotation by swapping axes order and flipping the data (resulting in 24 possible orientations); **random rotation**: Randomly rotates data in the $\mathrm{SO}(3)$ space.

As seen in Fig. 16 (a), all strategies enabled the model's loss to converge stably on the training set. However, Fig. 16 (b) and (c) indicate that the weakest augmentation strategy, center crop, leads to some overfitting on the test set. From the estimated likelihood of the data in the training and test set in Fig. 16 (d), model exhibits similar behavior across different strategies. From the downstream task performance shown in Fig. 16 (e), the differences among the strategies were negligible. Given that our modeling targets a more holistic $p(\mathbf{x})$, we choose random crop combined with random rotation as our augmentation strategy.

### H.2   PATCHIFYING & DOWNSAMPLING FACTOR

For three-dimensional data, the number of tokens is proportional to the cube of the edge length $D$ of the data. Applying patchification to the input and down-sampling within the model significantly im-

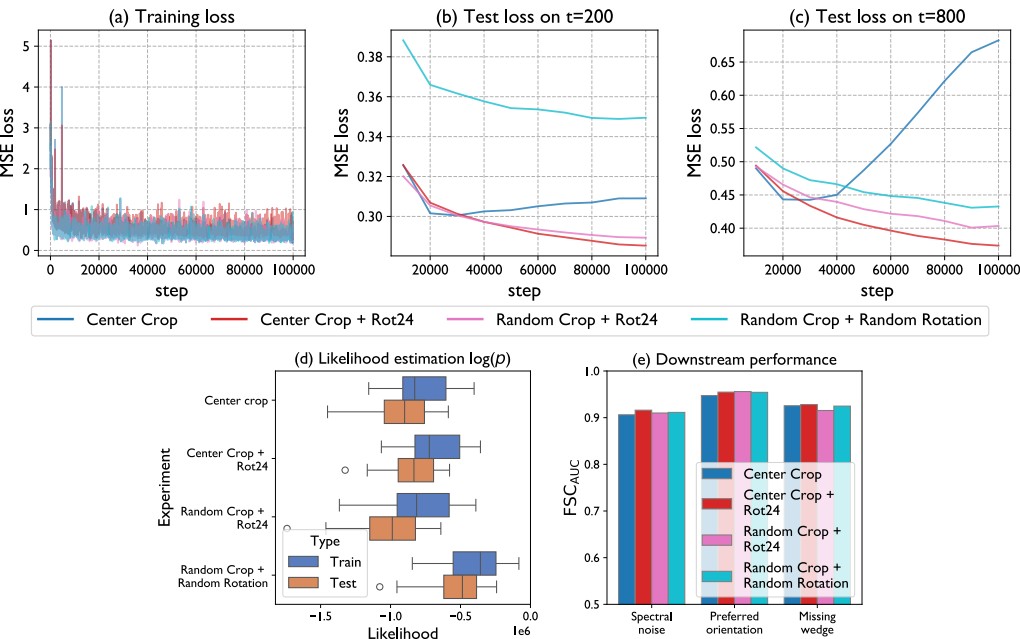

Figure 16: Results on different data augmentation strategies.

Table 7: Different model configurations.

|  | Default | Patchifying variants | | Down-sampling variants | |
| --- | --- | --- | --- | --- | --- |
| Short name | p4-d1 | p2-d1 | p8-d1 | p4-d0 | p4-d2 |
| Patching | 4 | 2 | 8 | 4 | 4 |
| Depth | [4, 8] | [4, 8] | [4, 8] | [12] | [2, 2, 8] |
| Width | [768, 1536] | [768, 1536] | [768, 1536] | [1536] | [384, 768, 1536] |
| Parameters | 335.18 M | 335.10 M | 335.87 M | 377.75 M | 316.06 M |
| Thru. (FPS) | 10.23 | 1.19 | 40.61 | 2.67 | 28.52 |

pacts the number of tokens the transformer block needs to process, thereby influencing the model's efficiency. In this section, we conducted ablation experiments on the number of patchifying and down-sampling operations. As shown in Tab. 7, the default configuration we used is p4-d1, where p4 represents a patchifying factor of 4, meaning a $4 \times 4 \times 4$ patch in the input is flattened and used as a token vector, and d1 represents that the data is down-sampled once in the model. We compare configurations with p = 2, 4, 8 and d = 0, 1, 2.

In Fig. 17, we present the loss curves on the test set during training for different model variants. Given the substantial differences in the computational costs among these model variants, the horizontal axis represents the training compute in Gflops. The estimation of training compute is: model Gflops · batch size · training steps · 3 (Peebles & Xie, 2023). We observe that the models with higher computational costs achieve lower loss levels, indicating the potential for scalability. On the other hand, as shown in Tab. 7, the inference speeds of the p2 and d0 models are exceedingly slow. Therefore, we selected the p4-d1 configuration with a comprehensive consideration of model performance and efficiency in downstream applications.

### H.3 HYPERPARAMETERS FOR POSTERIOR SAMPLING

Posterior sampling process involves continuously moving forward and correcting in the directions of prior $p(\mathbf{x})$ and the likelihood $p(\mathbf{x}|\mathbf{y})$. In this section, we present ablation experiments on the two parameters that significantly impact the effectiveness of the posterior sampling algorithm: the number of total timesteps, and $\lambda_{max}$ which controls the extent to which $\mathbf{x}_t$ is modified based on the observed data. We evaluated the performance using only the first four samples from the test set.

Tab. 8 and Tab. 9 present the ablation studies on these two parameters across three downstream tasks. The results show that for a fixed $\lambda_{max}$, increasing the number of sampling timesteps improves

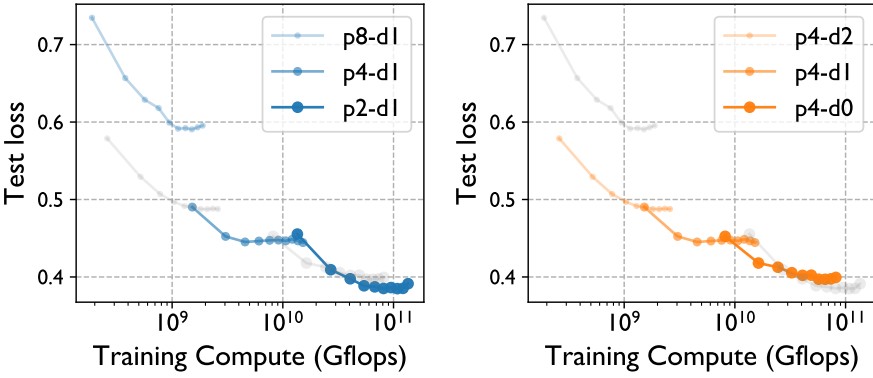

Figure 17: MSE loss on test set as a function of total training compute.

Table 8: Results with different posterior sampling parameters on spectral noise and anisotropic noise denoising task.

| | | Spectral Noise Denoising | | | | | Anistropic Noise Denoising | | |
|---|---|---|---|---|---|---|---|---|---|
| | | 3.2 Å | 4.3 Å | 6.1 Å | 8.5 Å | 15.0 Å | $\pm45°$ | $\pm30°$ | $\pm15°$ |
| After degradation | | 0.8874 | 0.5526 | 0.4968 | 0.3278 | 0.2000 | 0.6626 | 0.6326 | 0.6113 |
| $\lambda_{max}$ | # timesteps | | | | | | | | |
| 1.0 | 50 | 0.2511 | 0.2573 | 0.1811 | 0.1736 | 0.1349 | 0.1994 | 0.1638 | 0.1391 |
| 1.0 | 200 | 0.7301 | 0.5525 | 0.4985 | 0.3855 | 0.2476 | 0.5965 | 0.4836 | 0.4126 |
| 1.0 | 1000 | 0.9512 | 0.6771 | 0.6082 | 0.3928 | 0.2460 | 0.8745 | 0.8191 | 0.7741 |
| 5.0 | 50 | 0.4618 | 0.4140 | 0.3577 | 0.3140 | 0.2156 | 0.3778 | 0.2802 | 0.2576 |
| 5.0 | 200 | 0.8994 | 0.6417 | 0.5808 | **0.4115** | **0.2678** | 0.7923 | 0.6694 | 0.5714 |
| 5.0 | 1000 | **0.9545** | **0.6843** | **0.6191** | 0.3794 | 0.2313 | **0.8879** | **0.8453** | **0.8053** |
| 10.0 | 50 | 0.4989 | 0.4312 | 0.3904 | 0.3381 | 0.2249 | 0.4012 | 0.3027 | 0.2568 |
| 10.0 | 200 | 0.9025 | 0.6397 | 0.5798 | 0.4103 | 0.2631 | 0.7926 | 0.6713 | 0.5503 |
| 10.0 | 1000 | 0.9536 | 0.6649 | 0.6183 | 0.3752 | 0.2314 | 0.8780 | 0.8329 | 0.7926 |

quality, with the best performance at 1000 timesteps. Moreover, a $\lambda_{max}$ of 5 yields optimal or near-optimal results. Therefore, we defaulted to a $\lambda_{max}$ of 5 and 1000 timesteps for the posterior sampling parameters in our main results.

Table 9: Results with different sampling steps on missing wedge restoration task.

| | FSC$_{AUC}$ $\uparrow$ |
|---|---|
| After degradation | 0.7988 |
| Timesteps = 50 | 0.9295 |
| Timesteps = 200 | 0.9295 |
| Timesteps = 1000 | **0.9310** |

## H.4 LATENT FLOW MODELS

In this section, we discuss the choice of the input space of the flow-matching model: should we learn in the *voxel space* or the *latent space*? Most of the early works (Goodfellow et al., 2014; Van Den Oord et al., 2016; Brock, 2018; Dhariwal & Nichol, 2021) for generative modeling learned the data distribution directly on the original image space. Rombach et al. (2022) pointed out that learning diffusion models on the latent space can generate high-quality images with much less computational cost. The key is to learn an variational auto-encoder (VAE) (Kingma, 2014) which provides a low-dimensional representation that is equivalent to the original data space (Rombach et al., 2022; Podell et al., 2023). Then, a Diffusion Transformer (DiT) (Peebles & Xie, 2023) is trained on the latent codes of the VAE.

We have trained several latent flow models (LFMs). An LFM-$d$ model is comprised of a VAE-$d$ and a DiT, where $d$ denotes the downsampling factor of the VAE. A larger downsampling factor $d$ leads to greater information compression and increased information loss in the VAE, while it may ease the training of the DiT with smaller size of latent codes.

Fig. 11 shows the results of unconditional sampling from two latent flow models (LFM): LFM-4 and LFM-8, where the high-frequency information contains more noise, and secondary structures such as $\alpha$-helicies can not be observed. From the preliminary result, we concluded that: *For protein density modeling, the flow models trained in the voxel space converge much faster and better than the latent flow models.*

Our observation concides with that of Crowson et al. (2024), where they found that latent diffusion models may fail to generate fine details. We attribute the failure of latent flow models to the weak representational power of the learned variational auto-encoder (VAE) (Kingma, 2014). The protein density dataset for training the VAE is much smaller than the large-scale dataset in image synthesis tasks (Rombach et al., 2022; Podell et al., 2023). The low-dimensional space of the VAE is poorly regularized and can be sensitive to perturbations. A small error in the latent space may lead to large peceptual loss in the voxel space, especially for cryo-EM tasks which is highly sensitive to fine details.

### H.4.1 ARCHITECTURE OF LFM

It is somewhat difficult to conduct a rigorous comparison between latent-space models and voxel-space models as their architectures may differ in several aspects. For the completeness of this work, we will present the results of these models. However, for the sake of simplicity in presentation, we only show two representative settings, and the conclusion is generally consistent across various settings (such as parameter size, model architecture, batch size, etc.).

The procedure of learning a latent flow model whose downsampling factor is $d$ (LFM-$d$) can be divided into two parts:

- a VAE-$d$ which compresses a protein density $\mathbf{V} \in \mathbb{R}^{D \times D \times D}$ to a latent code $\mathbf{z} \in \mathbb{R}^{\frac{D}{d} \times \frac{D}{d} \times \frac{D}{d}}$, $d$ is the compressing factor. The compressing factor governs the balance between perceptual compression and distortion (Blau & Michaeli, 2018; Rombach et al., 2022). We employed the publicly available code of Stable Diffusion [6] by replacing 2D convolution layers with 3D layers. We conducted experiments with two settings: $d = 4$ and $d = 8$.

- a DiT (Diffusion Transformer) (Peebles & Xie, 2023) that learns the distribution of the latent codes. We experimented with a standard DiT architecture by replacing the 2D patchifying module with a 3D module [7]. The DiT has 12 layers, each of which has 1024 dimensions, and the patchify factor is set to 2.

Section H.4.2 shows that VAE-8 performs significantly worse in reconstruction tasks since it is over-compressed and causes loss of information. However, a larger downsampling factor produces smaller number of latent codes, which may ease the training of DiT. Therefore, we trained two DiTs on the latent codes of two VAEs. For each setting, we trained the DiT for 300k steps.

### H.4.2 EVALUATION OF VAE

Given a volume $\mathbf{V} \in \mathbb{R}^{D \times D \times D}$, we evaluate the reconstruction quality of VAE through auto-encoding Fourier shell correlation (AE-FSC) and auto-encoding L-1 loss between the input density and the reconstruction, defined by:

$$\text{AE-FSC} \triangleq \text{FSC}(\mathbf{V}, \text{VAE}(\mathbf{V}))$$

$$\text{AE-L1} \triangleq \|\mathbf{V} - \text{VAE}(\mathbf{V})\|_1$$

Tab. 10 shows that VAE-4 outperforms VAE-8 in terms of reconstruction quality, since VAE-8 learns to compress a grid of $8 \times 8 \times 8 = 512$ voxels into a latent code, which is 8 times harder than VAE-4 that operates on $4 \times 4 \times 4 = 64$ voxels.

---

[6]https://github.com/CompVis/stable-diffusion
[7]https://github.com/huggingface/diffusers

Table 10: Reconstruction quality of VAE

| Model | AE-FSC ($\uparrow$) | AE-L1 ($\downarrow$) | Training Steps |
|-------|---------------------|----------------------|----------------|
| VAE-4 | 0.9821 | 0.0028 | $588,480$ |
| VAE-8 | 0.8537 | 0.0049 | $1,097,872$ |

