# OpenReview forum: "CryoFM: A Flow-based Foundation Model for Cryo-EM Densities"
_ICLR.cc/2025/Conference — ICLR 2025 Poster_

### Official Review · Reviewer_esMq · 2024-10-28

**Soundness:** 3
**Presentation:** 3
**Contribution:** 3
**Rating:** 6
**Confidence:** 3

**Summary:**

The paper proposes a flow-based diffusion foundation model for 3D electron density maps of proteins as measured via cryogenic electron microscopy and tomography. This foundation model is then used as a prior for several 3D reconstruction and denoising tasks all framed as inverse problems: 3D density map denoising with spectral noise, 3D density map denoising with anisotropic noise, missing wedge restoration (for cryo-electron tomography), and ab initio reconstruction for cryo-electron microscopy from 2D projections.

**Strengths:**

There are many downstream protein-related modeling and reconstruction tasks that stand to benefit from an accurate model of the distribution of 3D protein density maps. The paper demonstrates applicability to several such tasks, and there are many more for which the proposed foundation model may be useful.

**Weaknesses:**

- The motivation for using a flow-based model is not very clear
- Related work discussion could be improved. I would suggest mentioning Chroma, a foundation model for protein structures that was published in 2023: https://www.nature.com/articles/s41586-023-06728-8. My understanding is that the main difference for the proposed model is that it operates in the space of 3D electron density maps rather than discrete protein models, yet for many cryo-EM tasks either or both models might be useful (e.g. https://arxiv.org/abs/2406.04239 this recent paper uses Chroma as a diffusion prior for several protein inverse problems). The authors might also consider discussing DiffModeler (https://www.nature.com/articles/s41592-024-02479-0), which also proposes a protein diffusion model that can denoise 3D density maps (preprint was posted in January 2024 but the published version was just released a week ago).
- Also pertaining to related work, the paper would benefit from a richer description of the specific baselines that are used for comparison in the experiments. These baselines are introduced in lines 116-120 but there is not enough detail to really judge what is similar and different about the mechanism of each model in the comparison.
- The experiments on anisotropic noise are not clearly motivated. It would seem to be somewhere in between the realistic noise models for cryoEM and cryoET, but both of these are separately considered so I don’t see much added value to consider anisotropic noise unless there is a measurement setting where it arises (that is not already covered by other experiments).
- There are specialized versions of the algorithm used for several of the applications, yet these modified algorithm descriptions are in the appendix rather than the main text. I realize space is limited but it would be preferable to have at least a description of the algorithm modifications in the main text.
- I am very happy to see experiments on ab initio reconstruction, but unfortunately the results in this setting are not all that compelling since the final quality is similar or slightly worse than CryoSPARC. Please refer to some questions about this experiment in the questions section.

Minor points
- The forward operator A is described as a “degradation operator”. In some cases this is accurate, such as when the measurements are noisy, but in other cases (such as ab initial reconstruction) my understanding is that A denotes projection rather than some form of degradation. Calling A a forward model/operator or measurement model/operator would be more consistent with the inverse problems literature.
- There is a typo on line 231; Transformer is misspelled.
- Figure 3: the colors for CryoFM and DeepEMhancer are too similar in the FSC curve. Another strategy would be to change the opacity or style of the lines so that both lines are visible even when they overlap.

**Questions:**

Most important questions:
- For the ab initio reconstruction experiments: (1) why not use many noisy projections instead of a few averaged/denoised projections? This would be more realistic to modern practice and more similar to CryoSPARC. (2) why not use CryoFM for refinement as well as initialization? Since you have a high-resolution foundation model I expect that you might see improvement by using it at the refinement stage.
- Do the authors plan to release their code and pretrained foundation model to the research community? My preference for paper acceptance assumes that these will be released.

Minor questions:
- Does the FM in the title stand for “foundation model”, “flow matching”, or both?
- Line 209-210 says that proteins with helical structures are removed from the pretraining dataset. Then line 348 says that the proposed method enhances “the resolution of helices and loops”…why were helical structures removed from the pretraining dataset if they will be important in the test dataset?
- What is the motivation behind the specific data augmentations that were used for CryoFM-S and CryoFM-L?
- Figure 9 does not show results for ab initio reconstruction; why?

---

> ### Author Response · Authors · 2024-11-21
>
> We thank the reviewer for the valuable questions. We have updated the manuscript and here is the detailed response:
>
> **Weakness 1: The motivation for using a flow-based model is not very clear**
>
> Thank you for your fruitful comments. The main motivation lies in the following two points: (i) generative modeling is a trend in constructing modern foundation models, and (ii) flow matching excels in generative modeling. Two representative foundation models (based on generative modeling) are AlphaFold3 and ESM3. Both of them incorporate an iterative refinement strategy to generate proteins (compared with their previous works, AlphaFold2 and ESM2). AlphaFold3 employs a diffusion module, and ESM3's refinement strategy resembles that of discrete diffusion. This indicates that the diffusion model has a great potential in building modern foundation models. Flow matching is a variant of diffusion models, and their training strategies are very similar. However, flow matching is easier to train and converges faster. We have added a discussion part in Appendix A.2 to discuss the motivation of using the flow matching framework for building foundation models.
>
> **W2: Related work discussion could be improved**
>
> Thank you for your helpful feedback.  We add the missing references and related discussions in the sections of introduction and related work. Besides that, (i) We add some discussion on foundation models in Section A.2 (ii) We originally intended to cite the DiffModeler in Section A.4 (which is Section B.4 in the updated manuscript), but we messed up the citation key and ended up citing another paper by mistake.
>
> **W3: More explanation of the baselines**
>
> Thank you for your suggestions. We have added more explanatory descriptions of these baselines in Appendix B.1.
>
> **W4: Motivation of anisotropic denoising**
>
> The anisotropic noise task is related to the preferred orientation problem, which is very common in single particle cryo-EM. During pose estimation, poses are often not uniformly distributed. That is, more particles may be assigned to a specific pose. Under a certain orientation, a density will have a higher signal-to-noise ratio if more particles are correctly assigned to these poses. This will make the final density have anisotropic noise instead of an isotropic spectral noise, which is the motivation of this task. From the measurement standpoint, this is very similar to the spectral denoising task, but from the application perspective, this addresses a more complicated problem that often happens in the real world.
>
> **W5: Specialized versions of the algorithm in the main text**
>
> We thank the reviewer for the fruitful suggestions. We have updated the manuscript and added some descriptions regarding the modifications. This undoubtedly makes our manuscript clearer.
>
> **Weakness minor points 1: ... “degradation operator” ... Calling A a forward model/operator or measurement model/operator would be more consistent with the inverse problems literature.**
>
> Thank you for your constructive suggestion regarding the terminology of the forward operator A. We appreciate your insight and have revised the manuscript to consistently use the term "forward model" instead of "degradation operator" to enhance consistency.
>
> **Weakness minor points 2: There is a typo on line 231; Transformer is misspelled.**
>
> Thanks, we have corrected the word.
>
> **Weakness minor points 3: Figure 3: the colors for CryoFM and DeepEMhancer are too similar in the FSC curve. Another strategy would be to change the opacity or style of the lines so that both lines are visible even when they overlap.**
>
> We thank the reviewer for the suggestion and have updated the figures to improve the clarity.

---

> ### Author Response · Authors · 2024-11-21
>
> **Q1: For ab initio: (1) why not use noisy particles? (2) why not use cryoFM as refinement as well?**
>
> (1) Our decision not to utilize real particles for ab initio reconstruction is based on several technical challenges that would significantly affect the accuracy and feasibility of the model:
> 1. **A large number of particles with low SNR complicate the sampling process**: Generally speaking, the time complexity of cryoFM for ab initio modeling is "sampling-step x projection-num". Dealing with a few 2D averages within the standard framework is considerably easier. However, when applied to tens of thousands of real particles, it necessitates numerous modifications to the algorithm to keep it robust. Moreover, the low SNR makes the pose estimation more arduous to implement, which may undermine the accuracy of likelihood computation. Conducting experiments on 2D averages with high SNR would be easier to validate the effectiveness of our model. In future work, we will refine our code for pose determination to be more robust.
> 2. **Inconsistent value range between the flow model and real particles**: When training cryoFM, we standardize the maps in the training set to ensure that their values range from 0 to 1. This is distinct from particles in the real world, the value of which may exhibit significant variance. Such a disparity makes it difficult to estimate the likelihood term since we cannot compute the MSE between two entities with different amplitudes. A workaround is to use correlation loss, which seems to alleviate the problem. However, we empirically find that due to the extremely low SNR of particles, the correlation loss may also fail.
> Given these challenges, we decided to use clean class averages for ab initio reconstruction and plan to focus on real particle reconstruction in future work. We also believe that incorporating CryoFM as a prior could potentially enhance the refinement process in addition to ab initio initialization. However, for the reasons outlined above, we will reserve this exploration for future work.
>
> (2) We agree with the reviewer and believe that CryoFM can be used for refinement as well. However, the details of the best way to incorporate CryoFM into refinement are not obvious and not in the scope of this paper, which we will leave for future work. For example, it seems feasible to embed CryoFM empowered denoising into each iteration in the refinement process, but this also introduces additional challenges, such as potential inaccuracies in estimating the forward model. We conducted additional experiments to evaluate the robustness of the estimated forward model, by introducing perturbations to the noise power of the estimated operator in the spectral denoising task. The results show that the model performs robustly with perturbations, suggesting that CryoFM has the potential to be applied to real data in refinement as well.
>
> Spectral noise denoising (reported numbers are $FSC_{AUC}$)
>
> |                     | 3.2 Å | 6.1 Å |
> |---------------------|---------|---------|
> | After degradation   | 0.8874  | 0.4970  |
> | w/o perturbation    | 0.9547  | 0.6188  |
> | w/ 5% perturbation  | 0.9547  | 0.6195  |
> | w/ 10% perturbation | 0.9546  | 0.6206  |
> | w/ 30% perturbation | 0.9541  | 0.6172  |
> | w/ 50% perturbation | 0.9527  | 0.6094  |
> | w/ 80% perturbation | 0.9480  | 0.5868  |

---

> > ### Author Response · Authors · 2024-11-21
> >
> > **Q2: ... release their code and pretrained foundation model ...**
> >
> > We plan to open-source our code and pretrained model upon acceptance.
> >
> > **Q3: Does the FM in the title stand for “foundation model”, “flow matching”, or both?**
> >
> > Thank you for your question. To keep things interesting and intriguing, we indeed intended FM to stand for both "foundation model" and "flow matching".
> >
> > **Q4: Line 209-210 "helical structures" vs line 348 "helices"**
> >
> > "Helical structures" in L209-210 refers to the densities that are multimers and exhibit a large scale helical feature, e.g. microtubles and amyloid-beta. We removed these entries from the training set because they are repetitive patterns that are extremely long, making training difficult. The helices in the "helices and loops" in line 348 refer to mostly alpha-helices, which are the main component of protein secondary structures.
> >
> > **Q5: Motivation behind the specific data augmentations for CryoFM-S and CryoFM-L**
> >
> > The only difference in data augmentation between CryoFM-S and CryoFM-L is that CryoFM-S uses random crop, while CryoFM-L uses center crop. CryoFM-S models high-resolution density maps with a voxel size of 1.5 Å/voxel. Due to GPU memory limitations, we restricted the input size to a $64\times64\times64$ box. Since 64x1.5 Å is insufficient to capture the entire protein, random crop was necessary. On the other hand, CryoFM-L models lower-resolution density maps with a voxel size of 3 Å/voxel and an input size of $128\times128\times128$. Most density maps fit within $128\times3$ Å, hence a center crop suffices to encompass the whole data.
> >
> > **Q6: Fig.9 why no ab initio?**
> >
> > Firstly, the metric for ab initio reconstruction is not $FSC_{AUC}$, making it unsuitable for inclusion in the same figure. Secondly, ab initio reconstruction and its evaluation require class averages and raw particles, which are usually in the EMPIAR dataset instead of EMDB. This makes it impossible to perform the ab initio reconstruction task on the 32 test cases in the other tasks.

---

> > > ### Comment · Reviewer_esMq · 2024-11-22
> > >
> > > I appreciate the answers to my questions, and maintain my score.

---

> > > > ### Author Response · Authors · 2024-11-25
> > > >
> > > > Thanks again for your suggestions which helps improve the quality of our manuscript!

---

### Official Review · Reviewer_xQqb · 2024-10-30

**Soundness:** 3
**Presentation:** 3
**Contribution:** 3
**Rating:** 6
**Confidence:** 4

**Summary:**

This study introduces CRYOFM, a foundation model using flow matching to learn and generalize the distribution of high-quality density maps. CRYOFM serves as a flexible prior for multiple tasks in cryo-EM and cryo-electron tomography (cryo-ET), achieving state-of-the-art performance across applications without fine-tuning, underscoring its potential for broad use in these fields.

**Strengths:**

1. The development of a foundation model for cryo-EM is both significant and novel
2. The use of flow matching to capture the prior distribution, along with the flow posterior sampling method as a flexible prior for various downstream tasks, is technically sound
3. Overall, the paper is well organized

**Weaknesses:**

1. While CRYOFM serves as a foundation model, it currently lacks demonstrated general applicability and versatility. As noted by the authors, further experiments across diverse tasks, such as 2D-to-3D reconstruction, would be beneficial to substantiate its broader utility.
2. CRYOFM relies on synthetic noise for denoising tasks. Testing on real datasets (realistic noise) would be necessary to validate its effectiveness in real-world applications.

**Questions:**

1. For the ab initio modeling approach, comparisons with established methods such as CryoDRGN2 and CryoFIRE would strengthen the evaluation of CRYOFM’s performance.
2. Regarding Figure 7, it would be interesting to see if using the same dataset with CryoSPARC actually improves performance.
3. Curious about effectiveness of the 2D to 3D reconstruction approach, particularly when only a limited set of 2D views is available

---

> ### Author Response · Authors · 2024-11-21
>
> We thank the reviewer for the valuable questions. We have updated the manuscript and here is the detailed response:
>
> **Weakness 1: ... lacks demonstrated general applicability and versatility ... such as 2D-to-3D reconstruction ...**
>
> Thank you for your insightful suggestions. We appreciate your recognition of the foundational aspect of cryoFM. Moving forward, we plan to investigate additional practical applications of cryoFM to demonstrate its broader utility further. Regarding the 2D-to-3D reconstruction, our current experiments focus primarily on using clean class averages as observations. We acknowledge that there are challenges when using experimental particle data directly. We conclude the challenges may come from the following reasons:
>
> 1. **A large number of particles with low SNR complicate the sampling process**: Generally speaking, the time complexity of cryoFM for ab initio modeling is "sampling-step x projection-num". Dealing with a few 2D averages within the standard framework is considerably easier. However, when applied to tens of thousands of real particles, it necessitates numerous modifications to the algorithm to keep it robust. Moreover, the low SNR makes the pose estimation more arduous to implement, which may undermine the accuracy of likelihood computation. Conducting experiments on 2D averages with high SNR would be easier to validate the effectiveness of our model. In future work, we will refine our code for pose determination to be more robust.
>
> 2. **Inconsistent value range between the flow model and real particles**: When training cryoFM, we standardize the maps in the training set to ensure that their values range from 0 to 1. This is distinct from particles in the real world, the value of which may exhibit significant variance. Such a disparity makes it difficult to estimate the likelihood term since we cannot compute the MSE between two entities with different amplitudes. A workaround is to use correlation loss, which seems to alleviate the problem. However, we empirically find that due to the extremely low SNR of particles, the correlation loss may also fail.
> However, we will reserve this exploration for future work.
>
> **W2: Testing on real datasets**
>
> The main reason that we did not use "experimental" noise for denoising is the lack of ground truth in this scenario, making result evaluation and comparison difficult. Although it is possible to apply cryoFM and other baselines to the density maps with a lower resolution, it is impossible to evaluate whether the authentic signals are enhanced and restored. Visual inspection can be deceptive and unreliable, as the seemingly improvement in the denoised density maps could simply be a hallucination. Dealing with experimental noise also introduces additional challenges, such as potential inaccuracies in estimating the forward model. On the other hand, spectral noise is a widely accepted noise model in cryo-EM reconstruction, and the anisotropic spectral noise is also reasonable when the pose distribution is uneven. Therefore, in order to provide a trustworthy metric evaluation compared to the ground truth, we adopted synthetic noise for denoising tasks.
>
> To further test the performance of cryoFM and flow posterior sampling in the case that the forward model may not be inferred accurately, we have conducted additional experiments to evaluate the robustness of the estimated forward model. We introduced perturbations to the noise power of the estimated operator in the spectral denoising task, and the experimental results indicate that the model performs robustly with the perturbations, showing that the model does not strictly rely on the precise estimate of the operator.
>
> Spectral noise denoising (reported numbers are $FSC_{AUC}$)
>
> |                     | 3.2 Å | 6.1 Å |
> |---------------------|---------|---------|
> | After degradation   | 0.8874  | 0.4970  |
> | w/o perturbation    | 0.9547  | 0.6188  |
> | w/ 5% perturbation  | 0.9547  | 0.6195  |
> | w/ 10% perturbation | 0.9546  | 0.6206  |
> | w/ 30% perturbation | 0.9541  | 0.6172  |
> | w/ 50% perturbation | 0.9527  | 0.6094  |
> | w/ 80% perturbation | 0.9480  | 0.5868  |

---

> > ### Author Response · Authors · 2024-11-21
> >
> > **Q1: ... comparisons with established methods such as CryoDRGN2 and CryoFIRE ...**
> >
> > We have conducted additional experiments comparing the performance of cryoFM with cryoDRGN2 and cryoFIRE. For cryoDRGN2 and cryoFIRE, the particles instead of the class averages were used as the input. The experiments were performed using the default parameters for both methods: 30 epochs for cryoDRGN2 and 200 epochs for cryoFIRE. However, due to the extended computational time required for cryoFIRE, we only run 100 epochs for EMPIAR-10827 (diminishing changes after 20 epochs). We present the comparative results in the table below, showing the resolution achieved in cryoSPARC homogeneous refinement by different ab initio reconstruction results:
> >
> > | Resolution                                 | EMPIAR-10028 | EMPIAR-10827 |
> > |--------------------------------------------|--------------|--------------|
> > | CryoSPARC ab initio + CryoSPARC refinement | 3.11 Å       | 4.75 Å       |
> > | CryoDRGN2 ab initio + CryoSPARC refinement | 3.10 Å       | 4.93 Å       |
> > | CryoFIRE ab initio + CryoSPARC refinement  | 8.73 Å       | 6.74 Å       |
> > | CryoFM ab initio + CryoSPARC refinement    | 3.15 Å       | 4.90 Å       |
> >
> > Our results indicate that cryoFM's performance is slightly worse than cryoSPARC ab-initio, yet comparable to cryoDRGN2. CryoFIRE struggled to produce favorable results without the ground truth (GT) pose (translation) provided.
> >
> > **Q2: Regarding Figure 7 ... if using the same dataset with CryoSPARC actually improves performance.**
> >
> > As mentioned in the feedback to Weakness 1, we explored using the same real particles in cryoFM but encountered challenges in achieving the desired outcomes. We guess the reasons behind may include: 1) A large number of particles with low SNR complicate the sampling process, 2) Inconsistent value range between the flow model and real particles, which makes the problem more complicated. We leave this for future work.
> >
> > **Q3: effectiveness of the 2D to 3D reconstruction approach, particularly when only a limited set of 2D views is available.**
> >
> > Thank you for raising this important point about the effectiveness of the 2D to 3D reconstruction approach, especially with limited 2D views available. Beyond the results we reported in the paper (including synthetic and real data), we have also tested additional cases using simulated data of different proteins. While one view is not enough to generate a reasonable result, two or three views are often sufficient. This aligns with our intuition, as successful reconstruction relies on diverse and informative views to capture the necessary information. For instance, as demonstrated with EMD-20795 in our paper, two distinct, informative views are already to adequately determine the protein's structure.

---

> > > ### Comment · Reviewer_xQqb · 2024-11-25
> > >
> > > I appreciate the authors answering my questions. I will maintain my score.

---

> > > > ### Author Response · Authors · 2024-11-26
> > > >
> > > > Thank you for taking the time to review our paper. We are glad that we were able to address your questions and concerns.

---

### Official Review · Reviewer_nPzD · 2024-10-30

**Soundness:** 3
**Presentation:** 2
**Contribution:** 3
**Rating:** 8
**Confidence:** 3

**Summary:**

This work is in the field of cryogenic electron microscopy (cryo-EM), a microscopy technique that can be used to study biological structures and macromolecules (e.g. proteins). Cryo-EM produces 2D projections of a structure, which are then used to estimate a 3D map, parametrized as a voxel array, representing the structure.

This paper addresses the problem of how to incorporate knowledge from the large database of existing high-quality 3D cryo-EM density maps into the process of estimating and refining new density maps from cryo-EM data. In the language of inverse problems, this problem translates to the question of how to obtain a prior for cryo-EM density maps.

The authors propose CryoFM, a generative deep learning model that captures the prior distribution of high-resolution cryo-EM density maps. After pre-training the model on a large dataset of cryo-EM density maps, the authors show that it can be used as a prior to regularize inverse problems where a 3D density map must be estimated from corrupted measurements of the underlying object. As examples of such inverse problems, the authors consider 3D denoising tasks and the problem of estimating the 3D density of an object from a few 2D projections of the density. To learn the prior from the data, the authors use techniques from flow matching (similar to denoising diffusion models).

**Recommendation**: In my opinion, the paper is above the acceptance threshold: Judging from the experiments, CryoFM is a good and versatile prior for cryo-EM, and the authors have done good work in the promising direction of learning a cryo-EM prior from data (see "Strengths")! *However*, parts of the paper should definitely be revised before acceptance. I find parts of the paper really hard to understand, and context and details are missing in some places (see "Weaknesses").
I will raise my score to "Accept" if my concerns related to the paper's clarity (see "Weaknesses") are sufficiently addressed.

**Post-Rebuttal:** I appreciate the authors' additional clarifications and increase my score from 6 to 8. Overall, the paper is an interesting case study of how to learn and use a prior for reconstruction and denoising in cryo-EM.
I also appreciate that the authors have made their training dataset publicly available (see "General Response"). I could not find the figshare link in the paper. I strongly encourage the authors to add the link to the paper, because I consider the dataset to be a valuable resource for future work in the direction of learning priors.

**Strengths:**

- The idea of learning a prior from data is promising. Work in other domains has shown that priors learned on large datasets can outperform classical, hand-crafted priors that, for example, model assumptions like smoothness or sparsity.
- The authors use the learned prior to solve four relevant inverse problems: two 3D denoising problems, the missing wedge problem (which deals with estimating missing data in the Fourier domain), and ab-initio modeling, which is the problem of estimating a 3D density from 2D projections of Cryo-EM.
- For all four inverse problems, the CryoFM-based approaches perform as well as or better than the baselines (see Table 1, Table 2, Figure 7), suggesting that CryoFM is a versatile and expressive prior.
- For ab-initio modeling, the authors tested CryoFM on two real-world datasets.
- All figures are of high quality.

**Weaknesses:**

In many places, the paper is hard to understand and details and context are lacking. For example:

- I find the theory of the flow-matching approach for training the prior very difficult to understand. This is especially true for Section 3 and Section 4.3, which together make up about 1.5 pages. In particular, important concepts and equations are not explained, making it difficult to understand their meaning. For example:
	- In lines 159 to 162, the authors state that the learnable vector field generates a probability density path, but I could not find an explanation of what such a path is or how the vector field relates to a probability density function.
	- I could not find a definition or explanation of the variable (or function?) $\hat{\mathbf{x}}_0$ (lines 186-187), which appears many times in the paper and is an important part of the flow posterior sampling (algorithm 1).

 - I cannot judge the novelty of the variant of flow matching used in the paper. If the variant contains novel elements, the authors should explain them and highlight the differences to existing work more clearly.

- Details and context are missing in some places in the experiments. For example:
	- The authors use EMReady, DeepEMhancer, and spIsoNet as baselines for the experiments in Section 5.1 and Section 5.2. They briefly introduce these baselines in lines 116 - 120, but I find this explanation insufficient to understand what the baselines actually do, why they were chosen, and how they relate to CryoFM.
	- In the denoising experiments in Sections 5.1 and 5.2, it is not clear what the test set is.
	- The motivation for studying anisotropic denoising (Section 5.2) is not entirely clear to me. Why does the noise affect "some orientations more than others"? Is this related to the preferred orientation problem, i.e. that some particles adopt certain orientations more often than others?

Other weaknesses:
- The authors propose a novel prior for solving inverse problems related to cryo-EM, but there is no discussion or experiment comparing CryoFM with existing priors, e.g. the Gaussian prior by Scheres et al. mentioned by the authors in line 38. One possible idea is to compare, for example, the Gaussian prior to CryoFM for regularizing ab-initio modeling.

- In Section 5.4, the authors use CryoFM for ab-initio modeling and compare it to cryoSPARC. It seems to me that cryoSPARC uses 2D projections of the particle directly, while CryoFM uses class averages. This difference in input data could be problematic when comparing the methods. This raises two questions:
	- Why did the authors use CryoSPARC as baseline?
	- Is there another ab-initio reconstruction method that works with class averages?

**Questions:**

- In Section 5.4, the authors show that CryoFM can be used for ab-initio modelling where a low-resolution density map is estimated from 2D projections. Can CryoFM also be used to estimate high resolution density maps from projections? Can CryoFM be for example incorporated in the "Refinement" step (done here with cryoSPARC)?


Note that I also raised some questions in the "Weaknesses" sections. They seemed more appropriate there.

---

> ### Author Response · Authors · 2024-11-21
>
> We thank the reviewer for the valuable questions. We have updated the manuscript and here is the detailed response:
>
> **Weakness 1: ... the theory of the flow-matching approach for training the prior very difficult to understand.**
>
> Thank you for your fruitful suggestions. We have updated the manuscript as follows: (i) Sections 3.1, 3.2, and 4.3 have been rewritten to make them more rigorous. (ii) A figure has been added in Appendix A.1 to illustrate the concepts of flow matching. (iii) A discussion part has been added in Appendix A.2 to discuss the motivation of using the flow matching framework for building foundation models.
>
> **Weakness 1: ... a definition or explanation of the variable (or function?) x^0 ...**
>
> Thank you for remarking that we missed some words for defining the function. $\hat{\mathbf{x}}_0(\mathbf{x}_t)$ represents the posterior mean of the flow (or the DDPM). To be more concise, it means "given a sample $\mathbf{x}_t$ at timestep $t$, what is the expected value at the final timestep?" It is not the output of the neural network $\mathbf{v} _\Theta$, but it can be computed analytically. For instance, $\hat{\mathbf{x}}_0(\mathbf{x}_t)=\mathbf{x}_t-t\cdot \mathbf{v} _\Theta(t, \mathbf{x}_t)$. We have added some words into the manuscript.
>
> **Weakness 2: Cannot judge the novelty of the variant of flow matching.**
>
> We thank the reviewer for the helpful feedback. In this manuscript, we did not alter the training framework of the original flow matching (Section 3.1). Our novelty lies in (i) exploring architectures and strategies (Section 4.1) for training flow matching models on 3D protein density data and (ii) following the DPS framework (Section 3.2) to derive a FPS framework (Section 4.3) and Algorithms 1/2/3 for solving specific inverse problems in cryo-EM.
>
>
> **Weakness 3 (Feedback 1~3): Details missing.**
>
> **Feedback 1: More explanation of the baselines**
>
> We thank the reviewer for this helpful feedback. We have added more explanatory descriptions of these baselines in the Appendix B.1.
>
> **Feedback 2: Test set in denoising experiments**
>
> The test set is 32 density maps which were selected and excluded from training. We have added a new section (C. Additional data collection and processing) in the Appendix discussing the curation of the data. And we have also uploaded the EMDB IDs of the training and test data used in this paper for readers to reference if needed https://figshare.com/s/9ef2614108391c04d910
>
> **Feedback 3: Motivation for anisotropic denoising**
>
> Yes, this is related to the preferred orientation problem. During pose estimation, poses are often not uniformly distributed. That is, more particles may be assigned to a specific pose. Under a certain orientation, a density will have a higher signal-to-noise ratio if more particles are correctly assigned to these poses. This will make the final density have anisotropic noise, which is the motivation of Section 5.2.

---

> > ### Author Response · Authors · 2024-11-21
> >
> > **W4: CryoFM vs Gaussian prior**
> >
> > The Gaussian prior introduced by Scheres et al. is a handcrafted prior to regularize that the reconstructed density map remains "smooth". This prior is modeled as a Gaussian distribution with zero mean, where the variance varies across different frequencies in Fourier space. During reconstruction, these frequency-dependent variances are inferred from the density map under an empirical Bayesian framework at each iteration during refinement [1]. For 3D reconstruction, the maximum a posteriori (MAP) solution has an analytical form (see Eq. 9 in [1]) that resembles a Wiener filter denoiser. Essentially, this Gaussian prior adjusts the amplitude of the reconstructed density based on the inferred signal-to-noise ratio (SNR) at each frequency. For instance, frequencies with high SNR (low frequency) are minimally altered, while those with low SNR (high frequency) are attenuated, promoting smoothness in the output density maps in a manner akin to low-pass filtering.
> >
> > In contrast, CryoFM introduces a deep, data-driven prior learned directly from a large dataset (EMDB), which provides a more informative and adaptive regularization. In the denoising tasks, CryoFM not only effectively suppresses noise but also enhances the authentic signal. This outperforms the Gaussian prior, which only applies a spectral "smoothing effect".
> >
> > [1] Scheres, Sjors HW. "A Bayesian view on cryo-EM structure determination." Journal of molecular biology 415.2 (2012): 406-418.
> >
> > **Weakness 5 (Feedback 1~3): CryoFM ab initio vs cryoSPARC; class averages vs particles**
> >
> > - **Feedback 1: why using cryoSPARC as the baseline?**
> >
> > To date, cryoSPARC is still the state-of-the-art method for ab initio reconstruction in single particle cryo-EM. Although the setting is different (using particles vs class averages), we believe that cryoSPARC is a strong and well recognized baseline for this task. Besides, we added experiments comparing the ab initio reconstruction using cryoDRGN2 and cryoFIRE.
> >
> > | Resolution                                 | EMPIAR-10028 | EMPIAR-10827 |
> > |--------------------------------------------|--------------|--------------|
> > | CryoSPARC ab initio + CryoSPARC refinement | 3.11 Å       | 4.75 Å       |
> > | CryoDRGN2 ab initio + CryoSPARC refinement | 3.10 Å       | 4.93 Å       |
> > | CryoFIRE ab initio + CryoSPARC refinement  | 8.73 Å       | 6.74 Å       |
> > | CryoFM ab initio + CryoSPARC refinement    | 3.15 Å       | 4.90 Å       |
> >
> > - **Feedback 2: Ab initio methods with class averages?**
> >
> > There are ab initio reconstruction methods that use class averages; however, these approaches are largely outdated due to the inherent difficulty of adequately filling Fourier space with only a limited number of projections [2]. We attempted to use e2initialmodel from EMAN2 for ab initio reconstruction with class averages, but the program failed to execute successfully.
> >
> > [2] Voss, Neil R., et al. "A toolbox for ab initio 3-D reconstructions in single-particle electron microscopy." Journal of structural biology 169.3 (2010): 389-398.

---

> > > ### Comment · Reviewer_nPzD · 2024-11-24
> > > **Score Increased**
> > >
> > > I thank the authors for their detailed responses. I have increased my score and updated my review.

---

> > > > ### Author Response · Authors · 2024-11-25
> > > >
> > > > We thank the reviewer for increasing the score, and your frutiful suggestions help a lot. We will add the link to the manuscript later.

---

### Official Review · Reviewer_a9QU · 2024-11-03

**Soundness:** 3
**Presentation:** 3
**Contribution:** 3
**Rating:** 6
**Confidence:** 4

**Summary:**

The paper introduces CryoFM, a foundation model for the distribution of high-quality cryo-EM density maps. CryoFM consists of two components: CryoFM-S models local details at high resolution, CryoFM-L models the overall shape at medium to low resolution. Both models are trained with data from EMDB by using flow matching. CryoFM can be combined with experimental data by using a flow posterior sampling method introduced by the authors. The impact of the foundation model is demonstrated for various reconstruction tasks (density modification, denoising, missing wedge problem, ab initio reconstruction) and in a number of ablation studies.

**Strengths:**

* CryoFM is a foundation model for density maps that can be combined with different types of data.
* CryoFM achieves state-of-the art performance on a number of reconstruction tasks and is demonstrated to be superior to other baselines.
* CryoFM seems to be faster than other baselines.

**Weaknesses:**

* The mathematical derivation of the algorithm seems to be a bit sloppy.
* Tests with experimental data are missing.
* No source code is provided (it would also help clarify some of the mathematical shortcomings of the paper).

__Recommendation__

I recommend a weak accept. The results are quite convincing, although a systematic benchmark with more examples would be desirable. It is unclear how the experimental density maps were selected for pretraining. It would be desirable if the authors reported a list of all PDB entries that went into pretraining. That way it is unclear whether some of the complexes that were used to illustrate the power of CryoFM are already part of the training data. Also the mathematical treatment of the method is sketchy and at times inconsistent.

**Questions:**

1) Dataset for pretraining: You state that you remove "problematic cases" from the pretraining set. What are these problematic cases? Why are large maps with a length greater than 576 \\AA removed from the pretraining set? Could you please provide a complete list of the maps used for pretraining?
2) You only show 4 examples for the ab initio modeling task, one sample for each problem setting. Can you comment on the success rate? Are the shown approximate posterior samples, the best human selected ones or something we can expect to see on average when using the proposed approach on other data?
3) Lines 375/376: From Fig. 5 it is unclear why and to what extent DeepEMhancer "introduces artifacts and alters the overall shape." The FSC is very good and the visual appearance is similar to your result.
4) How fast is CryoFM compared to the other baselines?

__Additional feedback__

1) The mathematical explanation of CryoFM should be improved. For example, Eq. (2) states $\\mathbf{x}_t | \\mathbf{x}_0 = (1-t)\\mathbf{x}_0 + t \\mathbf{x}_1$ implying $u(t, \\mathbf{x}_t | \\mathbf{x}_0) =  \frac{d}{dt} \\mathbf{x}_t|\\mathbf{x}_0 = \\mathbf{x}_1 - \\mathbf{x}_0$ and not $\\mathbf{x}_0 - \\mathbf{x}_1$. These sign errors occur at various places throughout the manuscript.

2) Another problem can be found in line 172 which should read
    $$\\mathcal{L} = \\mathbb{E}_{t, p_0(\\mathbf{x}_0), p_1(\\mathbf{x}_1)} \\left[ \\| v _{\\theta} (t, \\mathbf{x}_t) - (\\mathbf{x}_1 - \\mathbf{x}_0) \\|^2_2 \\right]$$
    In addition to the sign error, your loss is oversimplified and needs to indicate the expectation over $t \\sim \\mathcal{U}[0,1]$, $\\mathbf{x}_0 \\sim p_0(\\mathbf{x}_0)$, and  $\\mathbf{x}_1 \\sim p_1(\\mathbf{x}_1)$.

3) An error/typo can also be spotted in line 196:
    $$\\nabla_{\\mathbf{x}_t} \\log p (\\mathbf{x}_t | \\mathbf{y}) \\approx \\nabla _{\\mathbf{x}_t} \\log p (\\mathbf{x}_t) - \\lambda_t
\\nabla _{\\mathbf{x}_t}  \\| \\mathbf{y} - \\mathcal{A} \\hat{\\mathbf{x}}_0 (\\mathbf{x}_t) \\|_2^2$$
  Your version __adds__ the likelihood term rather than subtracting it, which only makes sense if $\\lambda_t \\leq 0$, but in Algorithms 1 and 3 we have $\lambda_t \\geq 0$.

4) ODE solvers require a time increment $\Delta t$ (explicitly or implicitly) which is missing in your version of Algorithm 1. So Algorithm 1 should read similar to

    \\begin{aligned}
    &\\mathbf{x}_1 \\leftarrow \\mathcal{N}(0, \\mathbf{I}) \\\\
    &\\textcolor{red}{\\Delta t \\leftarrow 1/N} \\\\
    &\\textbf{for }t \\in [1, 1- \\Delta t, 1 - 2\\Delta t, \\ldots, \\textcolor{red}{\\Delta t}] \\textbf{ do} \\\\
    &\\qquad \\mathbf{x}' _{t - \\textcolor{red}{\\Delta t}} \\leftarrow \\mathbf{x} _t  - v _{\\theta}(t, \\mathbf{x} _t) \\textcolor{red}{\\Delta t} \\\\
    &\\qquad l(\mathbf{x} _t) \\leftarrow \\| \\mathbf{y} - \\mathcal{A} \\hat{\\mathbf{x}} _0(\\mathbf{x} _t) \\|^2_2 \\\\
    &\\qquad \\mathbf{g} \\leftarrow \\frac{\\nabla _{\\mathbf{x} _t} l(\\mathbf{x} _t) }{ \\|\nabla _{\mathbf{x}_t } l( \mathbf{x}_t ) \\|_2} \\\\
    &\\qquad \\lambda _t \\leftarrow \\min \\{ \\lambda _{\\max}, t / (1-t) \\} \\\\
    &\\qquad \\mathbf{x} _{ t - \textcolor{red}{\\Delta t} } \\leftarrow  \\mathbf{x}' _{ t - \\textcolor{red}{\\Delta t} }  \\textcolor{red}{ - } \\lambda_t \\mathbf{g} \\\\
    & \\textbf{end for} \\\\
    & \\textbf{return } \\mathbf{x} _0
    \\end{aligned}

	Some additional comments on Algorithm 1:
	* What does $\\mathbf{x} _{t-1}$ mean? Is $t$ an index or the real-valued time $t \\in [0, 1]$? In $\\mathbf{x} _{t-1}$ it seems to be an index, in $\\mathbf{x}_0, \\mathbf{x}_1$ it indicates the time.
	* In your version, we have $t = 0$ in the last integration step such that $\\mathbf{x}_{-\\Delta t}$ is computed. But it does not make sense to decrease $t$ further after reaching the data distribution at $t = 0$. So the loop should end at $t = \\Delta t$.
	* Increasing the loss $l(\\mathbf{x}_t)$ does not make sense (most likely a typo or subsequent error of line 196).
	* In Algorithm 3, the same errors occur as in Algorithm 1.

5) Again, please clarify the meaning of $t-1$ in $\mathbf{x}_{t-1}$ and avoid division by 0 in $1/(Nt)$ in the (unnecessary) last iteration.

6) Symbol $\theta$ is overloaded: In most cases, $\theta$ indicates the parameters of the foundation model. But in the discussion of the missing wedge dataset it indicates the tilt angle.

7) It is confusing to use multiple versions of the flow. We see $v_t(\mathbf x_t)$, $v_\theta(t, \mathbf x_t)$ throughout the paper. Moreover, since the flow is a vector field, it should be bold face according to the ICLR guidelines.

# Minor comments

1) The specific number of 38,626 cryo-EM structures mentioned in the abstract will most likely be outdated at the time when your paper is published.

2) In line 095, you write "... which is then able to generate a posterior distribution $p(x|y)$." This is incorrect: Sampling with $v_t$ can only __approximate__ the posterior distribution.

3) Typos/grammar:
	* Page 7: "scaler"
	* Line 525: "We ... leave for future work."

---

> ### Author Response · Authors · 2024-11-21
>
> We thank the reviewer for the valuable questions. We would like to clarify that for tasks including spectral noise denoising, anisotropic noise denoising, and missing wedge restoration, all 32 cases in the test set were used to calculate the statistics of the metrics and reported in the tables in the paper.
>
> **Weakness 1 (and Feedback 1~8): the mathematical derivation of the algorithm seems to be a bit sloppy**
>
> There are mainly two concerns raised by the reviewer:
> - The sign error of vector field and the likelihood term in Section 3.2 and Section 4.3.
> - Some incorrect terms in the algorithms, including the missing time increment $\Delta t$, the confusing meaning of the subscript $t$.
> We find that all of your suggestions are correct and deeply appreciate the reviewer. We have updated the manuscript and hope it addresses your concerns. Here are the detailed responses:
>
> **Feedback 1: the sign error of the vector field**
> - We thank the reviewer for pointing this out. We changed the sign of the vector field in the updated version.
>
> **Feedback 2: the oversimplified loss**
> - Thank you for your feedback. We have updated the manuscript to make it more rigorous.
>
> **Feedback 3: the sign of the likelihood term**
> - This is our mistake. We are grateful for you pointing this out. The error arises from line 192 where the gradient to the log-probability should be proportional to the **negative** loss function. We have corrected both line 192 and line 196 in the updated manuscript.
> - the version in the original manuscript:
> $$\nabla_{\mathbf{x}_t} \log p(\mathbf{y}|\mathbf{x}_t) \propto \nabla _{\mathbf{x}_t} \\|\mathbf{y}-\mathcal{A}\hat{\mathbf{x}}_0(\mathbf{x}_t)\\|^2_2 $$
> - the correct version (updated):
> $$\nabla_{\mathbf{x}_t} \log p(\mathbf{y}|\mathbf{x}_t) \propto -\nabla _{\mathbf{x}_t} \\|\mathbf{y}-\mathcal{A}\hat{\mathbf{x}}_0(\mathbf{x}_t)\\|^2_2$$
>
> **Feedback 4: the missing explicit time increment in the algorithm**
> - Thank you! This makes the algorithms significantly more rigorous. We have updated Algorithm 1, Algorithm 2, and Algorithm 3.
>
> **Feedback 5.1 & Feedback 6: please clarify the meaning of $t-1$**
> - We are extremely grateful to the reviewer for bringing this to our attention. The subscript $t$ denotes a continuous time rather than a discrete index. We overlooked this when writing the manuscript, and we have updated all algorithms to enhance the clarity.
>
> **Feedback 5.2 & Feedback 6: the loop should end at $\Delta t$**
> - Thank you! You are correct and we have updated them.
>
> **Feedback 7: symbol $\theta$ is overloaded**
> - Thank you for your careful reading. We have changed the symbol of the network paramter from $\theta$ to $\Theta$. In cryo-EM, $\theta$ is widely adopted to denote a certain angle. Therefore, we change the symbol of the parameter.
>
> **Feedback 8: multiple versions of the flow, $\mathbf{v}_t(\mathbf{x}_t)$  and  $\mathbf{v} _\theta(t, \mathbf{x}_t)$, and the flow should be bold face**
> - On page 3 of the previous version, we wrote a footnote that "for ease of notation, sometimes we omit the $\theta$ and use $\mathbf{v}_t(\mathbf{x}_t)$ for abbreviation (of $\mathbf{v} _\theta(t, \mathbf{x}_t)$) in later sections." However, we concur with you that this is sometimes perplexing. Therefore, we have updated the manuscript to maintain consistency among all of them.
> - We have also changed all of the flow to bold face. Thank you for your careful reading!

---

> > ### Author Response · Authors · 2024-11-21
> >
> > **W2: Tests with experimental data are missing.**
> >
> > The main reason that we did not use "experimental" noise for denoising is the lack of ground truth in this scenario, making result evaluation and comparison difficult. Although it is possible to apply cryoFM and other baselines to the density maps with a lower resolution, it is impossible to evaluate whether the authentic signals are enhanced and restored. Visual inspection can be deceptive and unreliable, as the seemingly improvement in the denoised density maps could simply be a hallucination. Dealing with experimental noise also introduces additional challenges, such as potential inaccuracies in estimating the forward model. On the other hand, spectral noise is a widely accepted noise model in cryo-EM reconstruction, and the anisotropic spectral noise is also reasonable when the pose distribution is uneven. Therefore, in order to provide a trustworthy metric evaluation compared to the ground truth, we adopted synthetic noise for denoising tasks.
> > To further test the performance of cryoFM and flow posterior sampling in the case that the parameters in the forward model may not be estimated accurately, we have conducted additional experiments to evaluate the robustness of the estimated forward model. We introduced perturbations to the noise power of the estimated operator in the spectral denoising task, and the experimental results indicate that the model performs robustly with the perturbations, showing that the model does not strictly rely on the precise estimate of the operator.
> >
> > Spectral noise denoising (reported numbers are $FSC_{AUC}$)
> >
> > |                     | 3.2 Å | 6.1 Å |
> > |---------------------|---------|---------|
> > | After degradation   | 0.8874  | 0.4970  |
> > | w/o perturbation    | 0.9547  | 0.6188  |
> > | w/ 5% perturbation  | 0.9547  | 0.6195  |
> > | w/ 10% perturbation | 0.9546  | 0.6206  |
> > | w/ 30% perturbation | 0.9541  | 0.6172  |
> > | w/ 50% perturbation | 0.9527  | 0.6094  |
> > | w/ 80% perturbation | 0.9480  | 0.5868  |
> >
> > **W3: No source code is provided**
> > - We plan to further open-source our code and models upon acceptance.
> >
> > **Q1: Datasets for pretraining**
> >
> > We have uploaded our complete training and test EMDB id list to: https://figshare.com/s/9ef2614108391c04d910. For the "problematic cases", this includes density maps with very heterogeneous local resolutions that a significant portion of the map is poorly resolved. We added a new figure Fig. 13 in the Appendix to provide a few examples. We removed the very large maps because they occupy substantial disk space, creating an I/O bottleneck for model training.
> >
> > **Q2: ... Can you comment on the success rate? Are the shown approximate posterior samples, the best human selected ones or something we can expect to see on average when using the proposed approach on other data?**
> >
> > We thank the reviewer for the feedback. (i) We observe no significant difference between different runs and the result is relatively stable. (ii) In our early experiments, we find that picking 2D averages that can be under different views is beneficial, and this is easily identified by human eyes.
> >
> > **Q3: Lines 375/376: From Fig. 5 it is unclear why and to what extent DeepEMhancer "introduces artifacts and alters the overall shape." The FSC is very good and the visual appearance is similar to your result.**
> >
> > In Fig. 5, the FSC curve of DeepEMhancer (green) exhibits a sharp drop at low frequencies (around 20 Å resolution) compared to other methods. This decrease in correlation at low frequencies reflects a discrepancy in the overall shape relative to the ground truth. Specifically, the density produced by DeepEMhancer appears thinner in width compared to both the ground truth and the results of other methods.

---

> > > ### Author Response · Authors · 2024-11-21
> > >
> > > **Q4: How fast is CryoFM compared to the other baselines?**
> > >
> > > - We have summarized the inference time of CryoFM compared to the baselines. CryoFM is slower than the supervised pretrained models DeepEMhancer and EMReady, which only need to execute a single forward pass. However, CryoFM is significantly faster than spIsoNet, which requires retraining for each new dataset from scratch.
> > >
> > > |               | Elapsed time (per data) |
> > > |---------------|-------------------------|
> > > | DeepEMhancer  | ~1.5m                   |
> > > | EMReady       | ~0.5m                   |
> > > | spIsoNet      | ~3h45m                  |
> > > | CryoFM@200    | ~ 7m                    |
> > > | CryoFM@1000   | ~ 33m                   |
> > >
> > > - As a flow matching model, the number of sampling steps significantly impacts CryoFM's inference time. The results presented in the original paper were based on 1000 sampling steps. To provide a more comprehensive evaluation, we present results with 200 sampling steps (reported numbers are $FSC_{AUC}$):
> > >
> > > Spectral noise denoising
> > >
> > > |                   | 3.2 Å | 4.3 Å | 6.1 Å |
> > > |-------------------|-------|-------|-------|
> > > | After degradation | 0.89  | 0.55  | 0.50  |
> > > | CryoFM@200        | 0.89  | 0.63  | 0.57  |
> > > | CryoFM@1000       | 0.95  | 0.68  | 0.62  |
> > >
> > > Anistropic noise denoising
> > >
> > > |                   | ± 45° | ± 30° | ± 15° |
> > > |-------------------|-------|-------|-------|
> > > | After degradation | 0.66  | 0.63  | 0.61  |
> > > | CryoFM@200        | 0.78  | 0.68  | 0.59  |
> > > | CryoFM@1000       | 0.88  | 0.84  | 0.81  |
> > >
> > > Missing wedge restoration
> > >
> > > | After degradation | 0.80 |
> > > |-------------------|------|
> > > | CryoFM@200        | 0.92 |
> > > | CryoFM@1000       | 0.92 |
> > >
> > > - Our results indicate that reducing the sampling steps from 1000 to 200 offers a considerable reduction in inference time while maintaining competitive performance metrics in some settings. We have not yet applied some of the advanced strategies for accelerating sampling. Implementing these strategies in the future could substantially reduce the inference overhead and improve the overall efficiency of CryoFM.

---

> > > > ### Author Response · Authors · 2024-11-26
> > > >
> > > > Thank you again for your valuable feedback on our paper. We have updated both our responses and the manuscript accordingly. We hope that our revisions have resolved the concerns you mentioned.

---

### Author Response · Authors · 2024-11-21
**General Response**

We express our gratitude to all the reviewers for your valuable suggestions. We have addressed the questions with official comments under each individual review. We have updated the manuscript and conducted additional experiments. Here is a summary:

**(1) updated manuscript**:

- Fix typos in math equations and algorithms.
- Refine Section 3 and Section 4.3 to clarify the notation.
- Add a figure in Appendix A.1 to provide more details about the proposed algorithm.
- Add a discussion in Appendix A.2 on the motivation of flow matching.
- Add a discussion in Appendix B.1 on the related work.
- Add Appendix C for more details of data curation and preprocessing.
- Fix typos, confusing notations, and the misuse of terms.
- Add missing references.

**(2) List of EMDB entries for the training and test set**
We have added a new section (C. Additional data curation and processing) in the Appendix discussing the curation of the data. In Figure 13, we show examples of the entries we removed from the pretraining dataset after manual curation. Additionally, we have uploaded the EMDB IDs of the training and test data used in this paper for readers to reference if needed https://figshare.com/s/9ef2614108391c04d910 The 32 cases in the test set were for tasks including spectral noise denoising, anisotropic noise denoising, and missing wedge restoration. The results for these tasks, presented in the tables, represent statistics computed across all 32 test cases.

**(2) Real experimental noise for denoising**
The main reason that we did not use "experimental" noise for denoising is the lack of ground truth in this scenario, making result evaluation and comparison difficult. Although it is possible to apply cryoFM and other baselines to the density maps with a lower resolution, it is impossible to evaluate whether the authentic signals are enhanced and restored. Visual inspection can be deceptive and unreliable, as the seemingly improvement in the denoised density maps could simply be a hallucination. Dealing with experimental noise also introduces additional challenges, such as potential inaccuracies in estimating the forward model. On the other hand, spectral noise is a widely accepted noise model in cryo-EM reconstruction, and the anisotropic spectral noise is also reasonable when the pose distribution is uneven. Therefore, in order to provide a trustworthy metric evaluation compared to the ground truth, we adopted synthetic noise for denoising tasks.
To further test the performance of cryoFM and flow posterior sampling in the case that the forward model may not be inferred accurately, we have conducted additional experiments to evaluate the robustness of the estimated forward model. We introduced perturbations to the noise power of the estimated operator in the spectral denoising task, and the experimental results indicate that the model performs robustly with the perturbations, showing that the model does not strictly rely on the precise estimate of the operator.

Spectral noise denoising (reported numbers are $FSC_{AUC}$)

|                     | 3.2 Å | 6.1 Å |
|---------------------|---------|---------|
| After degradation   | 0.8874  | 0.4970  |
| w/o perturbation    | 0.9547  | 0.6188  |
| w/ 5% perturbation  | 0.9547  | 0.6195  |
| w/ 10% perturbation | 0.9546  | 0.6206  |
| w/ 30% perturbation | 0.9541  | 0.6172  |
| w/ 50% perturbation | 0.9527  | 0.6094  |
| w/ 80% perturbation | 0.9480  | 0.5868  |

**(4) Code availability**
We will open-source our code and models upon acceptance.

---

### Meta-Review · Area_Chair_VqgZ · 2024-12-19

**Metareview:**

The paper presents CryoFM, a generative model for cryoEM density maps. The model can be used to estimate a 3D density map from corrupted measurements. The paper considers 3D denoising tasks.

All four reviewers recommend acceptance, and note that the paper provides an effective approach to learn and use a prior for cryo-EM imaging.

The reviewers pointed to some weaknesses, in particular in the exposition, lack of context, lack of testing on real data, and a need to evaluate more diverse tasks. Most of those concerns were addressed in the rebuttal. For example, the authors improved clarity, and explain that they do not consider real-world data for a lack of ground-truth data. There are approaches to evaluate without ground-truth data, but the reconstruction problems considered in the paper are sufficiently interesting by itself. The reviewers see no major concerns after the rebuttal.

Some of the reviewers and I find the expositions of the model as a foundation model a bit misleading. The model is a generative model and the paper considers reconstruction tasks, thus it would be accurate to present the model and method as a reconstruction approach based on generative modeling, as opposed to a foundation model.

I recommend to accept the paper for its contribution to cyro-EM reconstruction.

**Additional Comments On Reviewer Discussion:**

see above

---

### Decision · Program_Chairs · 2025-01-22

Accept (Poster)